# An Information-Theoretic Perspective on Intrinsic Motivation in Reinforcement Learning: A Survey

**DOI:** 10.3390/e25020327

**Published:** 2023-02-10

**Authors:** Arthur Aubret, Laetitia Matignon, Salima Hassas

**Affiliations:** Univ Lyon, UCBL, CNRS, INSA Lyon, LIRIS, UMR5205, 69622 Villeurbanne, France

**Keywords:** intrinsic motivation, deep reinforcement learning, information theory, developmental learning

## Abstract

The reinforcement learning (RL) research area is very active, with an important number of new contributions, especially considering the emergent field of deep RL (DRL). However, a number of scientific and technical challenges still need to be resolved, among which we acknowledge the *ability to abstract actions* or *the difficulty to explore the environment in sparse-reward settings* which can be addressed by intrinsic motivation (IM). We propose to survey these research works through a new taxonomy based on information theory: we computationally revisit the notions of surprise, novelty, and skill-learning. This allows us to identify advantages and disadvantages of methods and exhibit current outlooks of research. Our analysis suggests that novelty and surprise can assist the building of a hierarchy of transferable skills which abstracts dynamics and makes the exploration process more robust.

## 1. Introduction

In reinforcement learning (RL), an agent learns by trial and error to maximize the expected rewards gathered as a result of their actions performed in the environment [1]. Traditionally, an agent maximizes a reward defined according to the task to perform it: this may be a score when the agent learns to solve a game, or a distance function when the agent learns to reach a goal. The reward is then considered as extrinsic (or as feedback) because the reward function is provided expertly, specifically for the task, and it is provided by an external entity. With an extrinsic reward, many spectacular results have been obtained on Atari games [2] with the deep q-network (DQN) [3] through the integration of deep learning to RL, leading to deep reinforcement learning (DRL).

However, despite the recent improvements of DRL approaches, they turn out to be unsuccessful most of the time when the rewards are scattered in the environment, as the agent is then unable to learn the desired behavior for the targeted task [4]. Moreover, the behaviors learned by the agent are rarely reusable, both within the same task and across many different tasks [4]. It is difficult for an agent to generalize the learned skills to make high-level decisions in the environment. Such a non-generalizable skill, for example, could be to *go to the door* using primitive actions consisting of moving in the four cardinal directions, or even to *move forward*, controlling different joints of a humanoid robot, such as in the robotic simulator MuJoCo [5].

On the other hand, unlike RL, developmental learning [6,7,8] is based on the idea that babies (or, more broadly, organisms) acquire new skills while spontaneously exploring their environment [9,10]. This is commonly called an intrinsic motivation (IM), which can be derived from an intrinsic reward. This kind of motivation allows one to autonomously gain new knowledge and skills, which then makes the learning process of new tasks easier [11]. For several years now, IM has been widely used in RL, fostered by important results and the emergence of deep learning. This paradigm offers a greater learning flexibility, through the use of a more general reward function, allowing the confrontation of the issues raised above when only an extrinsic reward is used. Typically, IM improves the agent’s ability to explore its environment, to incrementally learn skills independently of its main task, and to choose an adequate skill to be improved.

In this paper, we study and group together methods through a novel taxonomy based on information theoretic objectives. This way, we revisit the notions of surprise, novelty, and skill-learning, and show that they can encompass numerous works. Each class of the taxonomy is characterized by a computational objective that fits its eventual psychological definition. This allows us to relate to/situate a large body of works and to highlight important directions of research. To sum up, this paper investigates the use of IM in the framework of DRL and considers the following aspects:The role of IM in addressing the challenges of DRL.Classifying current heterogeneous works through a few information theoretic objectives.Exhibiting the advantages of each class of methods.Important outlooks of IM in RL within and across each category.

Related works

The overall literature on IM is huge [10]; we only consider its application to DRL and IMs related to information theory. Therefore, our study of IMs is not meant to be exhaustive. Intrinsic motivation currently attracts a lot of attention, and several works have made a restricted study of the approaches. Refs. [12] and [13], respectively, focus on the different aspects of skill-learning and exploration; Ref. [14] studies intrinsic motivation through the lens of psychology, biology, and robotics; Ref. [15] reviews hierarchical reinforcement learning as a whole, including extrinsic and intrinsic motivations; Ref. [16] experimentally compares different goal selection mechanisms. In contrast with these approaches, we study a large variety of objectives, all based on intrinsic motivation, through the lens of information theory. We assume that our work is in line with the work of [17], which postulates that organisms are guided by the desire to compress the information they receive. However, by reviewing the more recent advances in the domain, we formalize the idea of compression with the tools from information theory.

Some intrinsic motivations have been proposed based on the framework of information theory, but they do not fit the scope of this survey, as we decided to focus on single-agent exploration and skill-learning in reinforcement learning. For example, *empowerment* incites an agent to be in states from which it can reliably reach a large number of other states [18,19]. However, we are not aware of a successful use in complex DRL environments with respect to exploration. Some other approaches consider the synchronization of several agents [20,21,22,23]; are biased towards finding bottlenecks [24]; and model humans’ behaviors [25].

Structure of the paper

This paper is organized as follows. As a first step, we discuss RL, define intrinsic motivation and explain how it fits the RL framework (Section 2). Then, we highlight the main current challenges of RL and identify the need for an additional outcome (Section 3). Thereafter, we briefly explain our classification (Section 4), namely surprise, novelty, and skill learning, and we detail how current works fit it (respectively, Section 5, Section 6, and Section 7). Finally, we highlight some important outlooks of the domain (Section 8).

## 2. Definitions and Background

In this section, we will review the background of the RL field, explain the concept of IM, and describe how to integrate IM in the RL framework through goal-parameterized RL, hierarchical RL, and information theory. We sum up the notations used in the paper in Abbreviations.

### 2.1. Markov Decision Process

In RL, the problem is often modeled as the Markov decision process (MDP). The goal of a Markov decision process (MDP) is to maximize the expectation of cumulative rewards received through a sequence of interactions [26]. It is defined by *S*, the set of possible states; *A*, the set of possible actions; *T*, the transition function, $T:S\times A\times S\to p(s^{\prime }|s,a)$; *R*, the reward function, R:S×A×S→R; $d_{0}:S\to R$, the initial distribution of states. An agent starts in a state $s_{0}$, given by $d_{0}$. At each time step *t*, the agent is in a state $s_{t}$ and performs an action $a_{t}$, then it waits for feedback from the environment, composed of a state $s_{t+1}$ sampled from the transition function *T*, and a reward $r_{t}$ given by the reward function *R*. The agent repeats this interaction loop until the end of an episode. In reinforcement learning, the goal can be to maximize the expected discounted reward, defined by $\sum_{t=0}^{\infty }\gamma^{t}r_{t}$ where γ∈[0,1] is the discount factor. When the agent does not access the whole state, the MDP can be extended to a partially observable Markov decision process (POMDP) [27]. In comparison with a MDP, it adds a set of possible observations *O* which defines what the agent can perceive and an observation function Ω:S×O→R which defines the probability of observing o∈O when the agent is in the state *s*, i.e., Ω(s,o)=p(o|s).

A reinforcement learning algorithm aims to associate actions *a* to states *s* through a policy π. This policy induces a t-steps state distribution that can be recursively defined as:(1)$$d_{t}^{\pi }\left(S\right)=\int_{S}d_{t-1}^{\pi }\left(s_{t-1}\right)\int_{A}p\left(s_{t}|s_{t-1},a\right)\pi \left(a|s_{t-1}\right)da\;ds_{t-1}$$
with $d_{0}^{\pi }\left(S\right)=d_{0}$. The goal of the agent is then to find the optimal policy $\pi^{*}$ maximizing the reward:(2)$$\pi^{*}=\underset{\pi }{\arg\;\max}\;E_{\begin{array}{c} s_{0}∼d_{0}\left(S\right)\\ a_{t}∼\pi \left(\cdot |s{}_{t}\right)\\ s_{t+1}∼p\left(\cdot |s_{t},a_{t}\right) \end{array}}\left[\sum_{t=0}^{\infty }\gamma {}^{t}R\left(s{}_{t},a_{t},s_{t+1}\right)\right]$$
where x∼p(·) is equivalent to x∼p(x).

### 2.2. Definition of Intrinsic Motivation

Simply stated, intrinsic motivation is about doing something for its inherent satisfaction, rather than to obtain positive feedback from the environment [28]. In contrast, extrinsic motivation assumes there exists an expert or need that supervises the learning process.

According to [29], evolution provides a general intrinsic motivation (IM) function that maximizes a fitness function based on the survival of an individual. Curiosity, for instance, does not immediately produce selective advantages, but enables the acquisition of skills providing, by themselves, some selective advantages. More widely, the use of intrinsic motivation allows an agent to obtain intelligent behaviors which may later serve goals more efficiently than standard reinforcement [11,30,31]. Typically, a student doing his mathematical homework because they think it is interesting is intrinsically motivated, whereas their classmate doing it to get a good grade is extrinsically motivated [28]. In this example, the intrinsically motivated student may be more successful in math than the other one.

More rigorously, Ref. [32] explain that an activity *is intrinsically motivating for an autonomous entity if its interest depends primarily on the collation or comparison of information from different stimuli, independently of their semantics*. In contrast, an extrinsic reward results from an unknown environment static function which does not depend on the previous experience of the agent on the considered environment. The main point is that the agent must not have any a priori semantic knowledge of the observations it receives. Here, the term *stimuli* does not refer to sensory inputs, but more generally to the output of a system which may be internal or external to the independent entity, thereby including *homeostatic* body variables (temperature, hunger, thirst, attraction to sexual activities, etc.) [30].

Now that we clarified the notion of intrinsic motivation, we study how to integrate intrinsic motivation in the RL framework. An extensive overview of IM can be found in [10].

### 2.3. A Model of Rl with Intrinsic Rewards

Reinforcement learning usually uses extrinsic rewards [1]. However, Refs. [29,33] reformulated the RL framework to incorporate IM (Figure 1). We can differentiate *rewards*, which are events in the environment, and reward signals, which are internal stimulis to the agent. Thus, what is named reward in the RL community is in fact a reward signal. Inside the reward signal category, there is a distinction between primary reward signals and secondary reward signals. The secondary reward signal is a local reward signal computed through expected future rewards and is related to the value function, whereas the primary reward signal is the standard reward signal received from the MDP.

In addition, rather than considering the MDP environment as the environment in which the agent achieves its task, it suggests that the MDP environment can be formed of two parts: the external part, which corresponds to the potential task and the environment of the agent, and the internal part, which computes the MDP states and the secondary reward signal using potential previous interactions. Consequently, we can consider an intrinsic reward as a reward signal received from the MDP environment. The MDP state is no more the external state, but an internal state of the agent. However, from now, we will follow the terminology of RL and the term reward will refer to the primary reward signal.

Figure 1 summarizes the framework: the critic is in the internal part of the agent, it computes the intrinsic reward and deals with the credit assignment. The agent can merge intrinsic rewards and extrinsic rewards in its internal part. The main approach to integrate an intrinsic reward into a RL framework is to compute the reward of the agent *r* as a weighted sum of an intrinsic reward $r_{int}$ and an extrinsic reward $r_{ext}$: $r=\alpha r_{int}+\beta r_{ext}$ [34,35]. The state includes sensations and any form of internal context; in this section, we refer to this state as a contextual state. The decision can be a high-level decision decomposed by the internal environment into low-level actions.

This conceptual model incorporates intrinsic motivations into the formalism of MDP. Now, we will review how this model is instantiated in practice. Indeed, it is possible to extend RL to incorporate the three new components that are intrinsic rewards, high-level decisions, and contextual states. We study them separately in the following sections.

### 2.4. Intrinsic Rewards and Information Theory

Throughout our definition of intrinsic motivation, one can notice that the notion of *information* comes up a lot. Quantifying information often proves useful to generating intrinsic rewards. In this section, we provide the basics about information theory and explain how to combine intrinsic and extrinsic rewards.

The Shannon entropy quantifies the mean necessary information to determine the value of a random variable, or the disorder of a random variable. Let *X* be a random variable with a compact domain and a law of density p(X) satisfying the normalization and positivity requirements, we define its entropy by:(3)$$H\left(X\right)=-\int_{X}p\left(x\right)\log p\left(x\right)dx.$$
The entropy is maximal when *X* follows a uniform distribution, and minimal when p(X) is equal to zero everywhere except in one value, which is a Dirac distribution. We can also define the entropy conditioned on a random variable *S*. It quantifies the mean necessary information to find *X* knowing the value of another random variable *S*:(4)$$H\left(X|S\right)=-\int_{S}p\left(s\right)\int_{X}p\left(x|s\right)\log p\left(x|s\right)dxds.$$
The mutual information allows quantifying the information contained in a random variable *X* about another random variable *Y*, or the decrease of disorder brought by a random variable *Y* on a random variable *X*. The mutual information is defined by:(5)I(X;Y)=H(X)−H(X|Y).
We can notice that the mutual information between two independent variables is zero (since H(X|Y)=H(X)). The conditional mutual information allows quantifying the information contained in a random variable about another random variable, knowing the value of a third one. It can be written in various ways:
(6a)$$\begin{array}{cc} I(X;Y|S) & =H(X|S)-H(X|Y,S)=H(Y|S)-H(Y|X,S) \end{array}$$
(6b)$$\begin{array}{cc} \; & =D_{KL}\left[p(X,Y|S)||p(X|S)p(Y|S)\right] \end{array}$$
We can see with Equation ([Disp-formula FD6a-entropy-25-00327]) that the mutual information characterizes the decrease in entropy on X brought by Y (or inversely). Equation ([Disp-formula FD6b-entropy-25-00327]) defines the conditional mutual information as the Kullback–Leibler divergence [36], called $D_{KL}\left(\cdot ||\cdot \right)$, between distribution P(Y,X|S) and the same distribution if *Y* and *X* were independent variables (the case where H(Y|X,S)=H(Y|S)).

For further information on these notions, the interested reader can refer to [36]. Section 5, Section 6 and Section 7 illustrate how one can use information theory to reward an agent.

### 2.5. Decisions and Hierarchical RL

Hierarchical reinforcement learning (HRL) architectures are adequate candidates to model the decision hierarchy of an agent [37,38,39]. Initially, the hierarchical architectures have been introduced to make a long-term credit assignment easier [38,39]. This problem refers to the fact that rewards can occur with a temporal delay and will weakly affect all temporally distant states that have preceded them, although these states may be important for obtaining that reward. This problem also concerns determining which action is decisive for obtaining the reward, among all actions of the sequence. In contrast, if an agent can take advantage of temporally-extended actions, a large sequence of low-level actions become a short sequence of time-extended decisions that make the propagation of rewards easier.

Ref. [38] introduced the feudal hierarchy called *feudal reinforcement learning*. In this framework, a manager selects the goals that workers will try to achieve by selecting low-level actions. Once the worker has completed the goal, the manager can select another goal, so that the interactions keep going. The manager rewards the RL-based worker to guide its learning process; we formalize this with intrinsic motivation in the next section. This goal-setting mechanism can be extended to create managers of managers so that an agent can recursively define increasingly abstract decisions as the hierarchy of RL algorithms increases. Relatively to Figure 1, the internal environment of a RL module becomes the lower level module.

In the next section, we take a closer look at how to learn the policies that learn to achieve goals using intrinsic motivation. In particular, we will define goals, skills, and explain how to build a contextual state.

### 2.6. Goal-Parameterized Rl

Usually, RL agents solve only one task and are not suited to learning multiple tasks. Thus, an agent is unable to generalize across different variants of a task. For instance, if an agent learns to grasp a circular object, it will not be able to grasp a square object. In the developmental model described in Section 2.3, the decisions can be hierarchically organized into several levels, where an upper-level makes a decision (or sets goals) that a lower-level has to satisfy. This raises the questions of 1: how a DRL algorithm can make its policy dependent on the goal set by its upper-level decision module and 2: how to compute the intrinsic reward using the goal. These issues rise up a new formalism based on developmental machine learning [12].

In this formalism, a goal is defined by the pair $\left(g,R_{G}\right)$ where $G\subset R^{d}$, $R_{G}$ is a goal-conditioned reward function, and g∈G is the d-dimensional goal embedding. This contrasts with the notion of task, which is related to an extrinsic reward function assigned by an expert to the agent. With such embedding, one can generalize DRL to multi-goal learning with the universal value function approximator (UVFA) [40]. UVFA integrates, by concatenating, the goal embedding *g* with the state of the agent to create a contextual state c=(g,s). Depending on the semantic meaning of a skill, we can further enhance the contextual states with other actions or states executed after starting executing the skill (see Section 7).

We can now define the skill associated with each goal as the goal-conditioned policy $\pi^{g}\left(a|s\right)=\pi \left(a|g,s\right)$. This skill may be learned or unlearned according to the expected intrinsic rewards it creates. It implies that, if the goal space is well-constructed (often as a ground-state space, for example, $R_{G}=S$), the agent can generalize its neural network policy across the goal space, i.e., the corresponding skills of two close goals are similar. For example, let us consider an agent moving in a closed maze where every position in the maze can be a goal. We can set G=S and set the intrinsic reward function to be the Euclidean distance between the goal and the current state of the agent, $R_{G}:S\times G\to R,\left(s,g\right)\to ||s-{g||}_{2}$. Broadly speaking, the ability of a goal-conditioned policy to generalize and adapt to different situations (irrelevant objects or light intensity) [41] depends on how *g* and $R_{G}$ are built. Another alternative that does not benefit from such generalization can be to use expert goal-conditioned policies with independent learning parameters [41,42].

This formalism completes the production of the architectures described in Section 2.3.

## 3. Challenges of DRL

In this section, we detail two main challenges of current DRL methods that are partially addressed by IMs.

### 3.1. Sparse Rewards

Classic RL algorithms operate in environments where the rewards are dense, i.e., the agent receives a reward after almost every completed action. In this kind of environment, naive exploration policies such as ϵ-greedy [1] or the addition of a Gaussian noise on the action [43] are effective. More elaborated methods can also be used to promote exploration, such as Boltzmann exploration [3,44] or an exploration in the parameter space [45,46,47]. In environments with sparse rewards, the agent receives a reward signal only after it executes a large sequence of specific actions. The game *Montezuma’s revenge* [2] is a benchmark illustrating a typical sparse reward function. In this game, an agent has to move between different rooms while picking up objects (it can be keys to open doors, torches, etc.). The agent receives a reward only when it finds objects or when it reaches the exit of the room. Such environments with sparse rewards are almost impossible to solve with the above-mentioned *undirected* exploration policies [48] since the agent does not have local indications on the way to improve its policy. Thus, the agent never finds rewards and cannot learn a good policy with respect to the task [3]. Figure 2 illustrates the issue in a simple environment.

Rather than working on an exploration policy, it is common to shape an intermediary dense-reward function that adds to the reward associated to the task in order to make the learning process easier for the agent [49]. However, the building of a reward function often reveals several unexpected errors [50,51] and, most of the time, requires expert knowledge. For example, it may be difficult to shape a local reward for navigation tasks. Indeed, one has to be able to compute the shortest path between the agent and its goal, which is the same as solving the navigation problem. On the other hand, the automation of the shaping of the local reward (without calling on an expert) requires excessive computational resources [52]. We will see in Section 5, Section 6 and Section 7 how IM is a valuable method to encourage exploration when rewards are sparse.

### 3.2. Temporal Abstraction of Actions

As discussed in Section 2.5, skills, through hierarchical RL, can be a key element to speed up the learning process since the number of decisions to take is significantly reduced when skills are used. In particular, they make the *credit assignment* easier. Skills can be manually defined, but it requires some extra expert knowledge [39]. To avoid providing hand-made skills, several works proposed learning them with extrinsic rewards [53,54]. However, if an agent rather learns skills in a *bottom-up* way, i.e., with intrinsic rewards rather than extrinsic rewards, learned skills become independent of possible tasks. This way, skills can be reused across several tasks to improve transfer learning [42,55] and an agent can learn skills even though it does not access rewards, improving exploration when rewards are sparse [56]. Let us illustrate both advantages.

Figure 3 illustrates the benefit in terms of exploration when an agent hierarchically uses skills. The yellow agent can use the previously learned skill *Go to the far right* to reach the rewarding star, while the blue agent can only use low-level cardinal movements. The problem of exploration becomes trivial for the agent using skills, since one exploratory action can lead to the reward. In contrast, it requires an entire sequence of specific low-level actions for the other agent to find the reward. A thorough analysis of this aspect can be found in [57].

Skills learned with intrinsic rewards are not specific to a task. Assuming an agent is required to solve several tasks in a similar environment, i.e., a single MDP with a changing extrinsic reward function, an agent can execute its discovered skills to solve all tasks. Typically, in Figure 3, if both agents learned to reach the star, and we move the star somewhere else in the environment, the yellow agent would still be able to execute *Go to the far right*, and executing this skill may make the agent closer to the new star. In contrast, the blue agent would have to learn a whole new policy. In Section 7, we provide insights on how an agent can discover skills in a *bottom-up* way.

## 4. Classification of Methods

In order to tackle the problem of exploration, an agent may want to identify and return to rarely visited states or unexpected states, which can be quantified with current intrinsic motivations. We will particularly focus on two objectives that address the challenge of exploring with sparse rewards, each with different properties: maximizing novelty and surprise. Surprise and novelty are specific notions that have often been used in an interchanged way, and we are not aware of a currently unanimous definition of novelty [58]. The third notion we study, skill-learning, focuses on the issue of skill abstraction. In practice, surprise and novelty are currently maximized as flat intrinsic motivations, i.e., without using hierarchical decisions. This mostly helps to improve exploration when rewards are sparse. In contrast, skill-learning allows the definition of time-extended hierarchical skills that enjoy all the benefits argued in Section 3.2.

Table 1 sums up our taxonomy, based on information theory, which reflects the high-level studied concepts of novelty, surprise, and skill-learning. In practice, we mostly take advantage of the *mutual information* to provide a quantity for our conceptual objectives. These objectives are compatible with each other and may be used simultaneously, as argued in Section 8.3. Within each category of objectives, we additionally highlight several ways to maximize each objective and provide details about the underlying methods of the literature. We sum up surprise, novelty, and skill-learning methods, respectively, in Table 2, Table 3 and Table 4.

While a full benchmark of the studied methods on several environments is beyond the scope of the survey, we provide some performance metrics to give a evidence about how each method compares to the others. In Table 2 and Table 3, we compare some methods according to their performance on the sparse reward environment *Montezuma’s revenge* when available, as 1: it is a sparse-reward benchmark widely used to assess the ability of the method to explore; 2: unlike most IMs, a standard DQN achieves a score of near 0 in this environment [3]; 3: the *Montezuma’s revenge* score directly reflects the number of rooms visited by the agent; 4: the score correlates with human behaviors [59]. In Table 4, we also compare skill-learning methods from Section 7.2 and Section 7.3 according to their performance on the widely used hierarchical task of the *ant maze*, and question whether they need a hand-made goal space (x, y) or an implicit curriculum of objectives. For methods in Section 7.1, we did not find a representative or widely used evaluation protocol/environment among the published works.

## 5. Surprise

In this section, we study methods that maximize surprise. Firstly, we formalize the notions of surprise, then we will study three approaches for computing intrinsic rewards based on these notions.

### 5.1. Surprise Maximization

In this section, we assume the agent learns a forward model of the environment (Section 5.4 and Section 5.2) parameterized by ϕ∈Φ. A forward model computes the next-state distribution conditioned on a tuple state-action $p(S^{\prime }|S,A,\phi )$. Typically, this is the parameters of a neural network. Trying to approximate the true model, the agent maintains an approximate distribution p(Φ|h) of models, where $h_{t}=h$ refers to the ordered history of interactions $\left(\left(s_{0},a_{0},s_{1}\right),\left(s_{1},a_{1},s_{2}\right),\cdots ,\left(s_{t-1},a_{t-1},s_{t}\right)\right)$. *h* simulates a dataset of interactions fed with the policy, and we use it to clarify the role of the dataset. It induces that the agent has a prior expectation about the model parameters’ distribution p(Φ) and this model can be updated using the Bayes rule:(7)$$p\left(\phi |h,s,a,s^{\prime }\right)=\frac{p\left(\phi |h\right)\;p\left(s^{\prime }|h,s,a,\phi \right)}{p(s^{\prime }|h,s,a)}.$$

Surprise usually quantifies the mismatch between an expectation and the true experience of an agent [58,60]. Here, we instead propose a more general formalism which, as argued in the next sections, accounts for several intrinsic motivations. We assume that there is a distribution of true models $p(\Phi_{T})$ that underpins the transition function of the environment *T*. In contrast with Φ, this is a property of the environment. One can see this distribution is a Dirac distribution if only one model exists, or as a categorical distribution of several forward models. We consider an agent that maximizes the expected information gain over the true models as:
(8a)$$\begin{array}{cc} IG(h,A,S^{\prime },S,\Phi_{T}) & =I\left(S^{\prime };\Phi_{T}|h,A,S\right)=H\left(\Phi_{T}|h,A,S\right)-H\left(\Phi_{T}|h,A,S,S^{\prime }\right) \end{array}$$
(8b)$$\begin{array}{cc} \; & =E_{\begin{array}{c} \left(s,a\right)∼p\left(\cdot |h\right),\;\phi_{T}∼p\left(\cdot \right)\\ s^{\prime }∼p\left(\cdot |s,a,\phi_{T}\right) \end{array}}\log p\left(s^{\prime }|s,a,h,\phi_{T}\right)-\log p\left(s^{\prime }|s,a,h\right) \end{array}$$
Maximizing Equation ([Disp-formula FD8b-entropy-25-00327]) amounts to looking for states that inform about the true models’ distribution. We can see that the left-hand side of Equation ([Disp-formula FD8b-entropy-25-00327]) incites the agent to target inherently deterministic areas, i.e., given the true forward model, the agent would know exactly where it ends up. Conversely, the right-hand term pushes the agent to move in stochastic areas according to its current knowledge. Overall, to improve this objective, an agent has to reach areas that are more deterministic than what it thinks they are. In practice, this objective is not directly optimizable since the agent does not access the true forward model. In this perspective, we propose that surprise results from an agent-centric approximation of Equation ([Disp-formula FD8b-entropy-25-00327]) based on an internal model.

In the following, we will study three objectives: the prediction error, the expected information gain over the forward model, and the expected information gain over the density models.

### 5.2. Prediction Error

To avoid the need for the true forward model, the agent can omit the left-hand term of Equation ([Disp-formula FD8b-entropy-25-00327]) by assuming the true forward model is modeled as a deterministic forward model. In this case, we can write:
(9a)$$\begin{array}{cc} I(S^{\prime };\Phi_{T}|h,A,S) & \propto E_{\begin{array}{c} \left(s,a\right)∼p\left(\cdot |h\right),\phi_{T}∼p\left(\cdot \right)\\ s^{\prime }∼p\left(\cdot |s,a,\phi_{T}\right) \end{array}}-\log p\left(s^{\prime }|s,a,h\right) \end{array}$$
(9b)$$\begin{array}{cc} \; & =E_{\begin{array}{c} \left(s,a\right)∼p\left(\cdot |h\right),\;\phi_{T}∼p\left(\cdot \right)\\ s^{\prime }∼p\left(\cdot |s,a,\phi_{T}\right) \end{array}}-\log\sum_{\phi \in \Phi }p\left(s^{\prime }|h,s,a,\phi \right)p\left(\phi |h\right) \end{array}$$
(9c)$$\begin{array}{cc} \; & \geq E_{\begin{array}{c} \phi_{T}∼p\left(\cdot \right),\;\left(s,a\right)∼p\left(\cdot |h\right)\\ s^{\prime }∼p\left(\cdot |s,a,\phi_{T}\right),\phi ∼p\left(\cdot |h\right) \end{array}}-\log p\left(s^{\prime }|h,s,a,\phi \right) \end{array}$$
where we applied the Jensen inequality in Equation ([Disp-formula FD9c-entropy-25-00327]) and $\phi_{T}$∼p(·) is fixed. One can model $p(s^{\prime }|h,s,a,\phi )$ with a unit-variance Gaussian distribution in order to obtain a simple and tractable loss. Empirical results suggest this distribution works well even with high-dimensional inputs [61]. This way, we have:
(10a)$$\begin{array}{ll} E_{\begin{array}{c} \left(s,a\right)∼p\left(\cdot |h\right),\;\phi_{T}∼p\left(\cdot \right)\\ s^{\prime }∼p\left(\cdot |s,a,\phi_{T}\right),\;\phi ∼p\left(\cdot |h\right) \end{array}}- & \log p(s^{\prime }|\phi ,h,a,s)\\ & \approx E_{\begin{array}{c} \left(s,a\right)∼p\left(\cdot |h\right),\;s^{\prime }∼p\left(\cdot |s,a,\phi_{T}\right)\\ \phi ∼p\left(\cdot |h\right),\;\phi_{T}∼p\left(\cdot \right) \end{array}}-\log\frac{1}{{\left(2\pi \right)}^{d/2}}e^{-0.5{\left(s^{\prime }-{\hat{s}}^{\prime }\right)}^{T}\left(s^{\prime }-{\hat{s}}^{\prime }\right)} \end{array}$$
(10b)$$\begin{array}{cc} \; & \propto E_{\begin{array}{c} \left(s,a\right)∼p\left(\cdot |h\right),\;s^{\prime }∼p\left(\cdot |s,a,\phi_{T}\right)\\ \phi ∼p\left(\cdot |h\right),\;\phi_{T}∼p\left(\cdot \right) \end{array}}\left|\right|s^{\prime }-{\hat{s}}^{\prime }{\left|\right|}_{2}^{2}+Const \end{array}$$
where
(11)$${\hat{s}}^{\prime }=\underset{s^{\prime \prime }\in S}{\arg\;\max}\;p\left(s^{\prime \prime }|h,a,s,\phi \right)$$
represents the mean prediction and ϕ parameterizes a deterministic forward model. Following the objective, we can extract a generic intrinsic reward as:(12)$$\begin{array}{c} R\left(s,a,s^{\prime }\right)=||f\left(s^{\prime }\right)-f\left({\hat{s}}^{\prime }\right){\left|\right|}_{2}^{2} \end{array}$$
where *f* is a generic function (e.g., identity or a learned one) encoding the state space into a feature space. Equation (Equation 12) amounts to rewarding the predictor error of ϕ in the representation *f*. In the following, we will see that learning a relevant function *f* is the main challenge.

The first natural idea to test is whether a function *f* is required. Ref. [62] learns the forward model from the ground-state space and observes that it is inefficient when the state space is large. In fact, the Euclidean distance is meaningless in such high-dimensional state space. In contrast, they raise that random features extracted from a random neural network can be very competitive with other state-of-the-art methods. However, they poorly generalize to environment changes. Another model, the *dynamic auto-encoder (dynamic-AE)* [63], selects *f* to be an auto-encoder [64], but it only slightly improves the results over the Boltzmann exploration on some standard Atari games. Other works also consider a dynamic-aware representation [65]. These methods are unable to handle the local stochasticity of the environment [62]. For example, it turns out that adding random noise in a 3D environment attracts the agent; it passively watches the noise since it is unable to predict the next observation. This problem is also called *the white-noise* problem [66,67]. This problem emerges by considering only the right-hand term of Equation ([Disp-formula FD8b-entropy-25-00327]), making the agent assume that environments are deterministic. Therefore, exploration with prediction error breaks down when this assumption is no longer true.

To tackle exploration with local stochasticity, the *intrinsic curiosity module (ICM)* [66] learns a state representation function *f* end to end with an *inverse model*, i.e., a model which predicts the action between two states. Thus, the function *f* is constrained to represent things that can be controlled by the agent during the next transitions. The prediction error does not incorporate the white noise does not depend on actions, so it will not be represented in the feature state space. Building a similar action space, *exploration with mutual information (EMI)* [68] significantly outperforms previous works on Atari, but at the cost of several complex layers. EMI transfers the complexity of learning a forward model into the learning of states and actions representation through the maximization of $I(\left[S,A\right];S^{\prime })$ and $I(\left[S,S^{\prime }\right];A)$. Then, the forward model ϕ is constrained to be a simple linear model in the representation space. Furthermore, EMI introduces a *model error* which offloads the linear model when a transition remains strongly non-linear (such as a screen change). However, it is currently unclear whether ICM and EMI keep in their representation what depends on their long-term control. For instance, in a partially observable environment [27], an agent may perceive the consequences of its actions several steps later. In addition, they remain sensitive to stochasticity when it is produced by an action [62,69].

### 5.3. Learning Progress

Another way to tackle local stochasticity can be to maximize the improvement of prediction error, or learning progress, of a forward model [70,71,72,73,74] or local forwards models [75,76] One can see this as approximating the left-hand side of Equation ([Disp-formula FD8b-entropy-25-00327]) with:(13)$$\begin{array}{cc} \log p\left(s^{\prime }|s,a,h,\phi_{T}\right)-\log p\left(s^{\prime }|s,a,h\right) & \approx \log p\left(s^{\prime }|s,a,h^{\prime }\right)-\log p\left(s^{\prime }|s,a,h\right) \end{array}$$(14)$$\begin{array}{cc} \; & \approx \Delta ||f\left(s^{\prime }\right)-f\left({\hat{s}}^{\prime }\right){\left|\right|}_{2}^{2} \end{array}$$
where $h^{\prime }$ concatenates *h* with an arbitrary number of additional interactions and where we derive Equation (Equation 14) similarly to Equation (Equation 12). As $h^{\prime }$ becomes large enough and the agent updates its forward model, its forward model converges to the true transition model. Formally, if one stochastic forward model can describe the transitions, we can write:(15)$$\begin{array}{cc} \lim_{|h^{\prime }|\to \inf}p\left(s^{\prime }|s,a,h^{\prime }\right) & =\lim_{|h^{\prime }|\to \inf}\sum_{\Phi }p\left(s^{\prime }|s,a,h^{\prime },\phi \right)p\left(\phi |h^{\prime }\right)\\ \; & =p(s^{\prime }|s,a,h^{\prime },\phi_{T}) \end{array}$$
In practice, one cannot wait to discover a long sequence of new interactions and the reward can be dependent on a small set of interactions and the efficiency of the gradient update of the forward model. It makes it hard to compute a tractable reward of learning progress in complex environments. We refer to [16] for a study of learning progress measures. However, the theoretical connection with the true expected information gain may indeed explain the robustness of learning progress to stochasticity [16].

### 5.4. Information Gain over Forward Model

By assuming $p\left(s^{\prime }|s,a,h,\phi_{T}\right)\approx p\left(s^{\prime }|s,a,\phi ,h\right)$, we can approximate the expected information gain over true models by the expected information gain [77,78]:
(16a)$$\begin{array}{cc} IG(h,A,S^{\prime },S,\Phi_{T}) & =I\left(S^{\prime };\Phi_{T}|h,A,S\right)=E_{\begin{array}{c} \left(s,a)∼p(\cdot |h\right)\\ s^{\prime }∼p\left(\cdot |s,a,h,\phi_{T}\right) \end{array}}D_{KL}(p\left(\Phi_{T}|h,s,a,s^{\prime }\right)||p\left(\Phi_{T}|h\right)) \end{array}$$
(16b)$$\begin{array}{cc} \; & \approx E_{\begin{array}{c} (s,a)∼\pi \\ s^{\prime }∼p\left(\cdot |s,a,h,\phi_{T}\right) \end{array}}D_{KL}(p\left(\Phi |h,s,a,s^{\prime }\right)||p\left(\Phi |h\right)). \end{array}$$
In Equation ([Disp-formula FD16b-entropy-25-00327]), the agents acts and uses the new states to update its own model. The amplitude of the update defines the reward
(17)$$R\left(s,a,s^{\prime }\right)=D_{KL}(p\left(\Phi |h,s,a,s^{\prime }\right)||p\left(\Phi |h\right)).$$
This way, an agent executes actions that provide information about the dynamics of the environment. This allows, on one side, pushing the agent towards areas it does not know, and on the other side, prevents attraction towards stochastic areas. Indeed, if the area is deterministic, environment transitions are predictable and the uncertainty about its dynamics can decrease. Conversely, if transitions are stochastic, the agent turns out to be unable to predict transitions and does not reduce uncertainty. The exploration strategy *VIME* [79] computes this intrinsic reward by modeling p(ϕ|h) with Bayesian neural networks [80]. The interest of Bayesian approaches is to be able to measure the uncertainty of the learned model [81]. This way, assuming a fully factorized Gaussian distribution over model parameters, the KL-divergence has a simple analytic form [16,79], making it easy to compute. However, the interest of the proposed algorithm is shown only on simple environments and the reward can be computationally expensive to compute. Ref. [82] proposes a similar method (*AKL*) with comparable results, using deterministic neural networks, which are simpler and quicker to apply. The weak performance of both models is probably due to the difficulty of retrieving the uncertainty reduction by rigorously following the mathematical formalism of information gain.

The expected information gain can also be written as
(18a)$$\begin{array}{cc} I & \left(S^{\prime };\Phi |h,A,S\right)=H\left(S^{\prime }|h,A,S\right)-H\left(S^{\prime }|h,A,S,\Phi \right)\\ & =-E_{\begin{array}{c} (s,a)∼p(\cdot |h),\\ s^{\prime }∼p\left(\cdot |h,a,s,\phi_{T}\right) \end{array}}\log p\left(s^{\prime }|h,a,s\right)+E_{\begin{array}{c} \phi ∼p\left(\cdot |h,a,s,s^{\prime }\right),\;\left(s,a\right)∼p\left(\cdot |h\right)\\ s^{\prime }∼p\left(\cdot |s,a,h,\phi_{T}\right) \end{array}}\log p\left(s^{\prime }|h,a,s,\phi \right) \end{array}$$
(18b)$$\begin{array}{cc} \; & =E_{\begin{array}{c} \phi ∼p(\cdot |h,a,s,s^{\prime }),\\ \left(s,a\right)∼p\left(\cdot |h\right)s^{\prime }∼p\left(\cdot |s,a,h,\phi_{T}\right) \end{array}}-\log\sum_{\phi \in \Phi }p\left(s^{\prime }|h,a,s,\phi \right)p\left(\phi |h\right)+\log p\left(s^{\prime }|h,a,s,\phi \right). \end{array}$$
Using similar equations as in Equation ([Disp-formula FD18b-entropy-25-00327]), in *JDRX* [83], authors show that one can maximize the information gain by computing the Jensen–Shannon or Jensen–Rényi divergence between distributions of the next states induced by several forward models. The more the different models are trained on a state-action tuple, the more they will converge to the expected distribution of the next states. Intuitively, the reward represents how much the different transition models disagree on the next-state distribution. Other works also maximize a similar form of disagreement [84,85,86] by looking at the variance of predictions among several learned transition models. While these models handle the white-noise problem, the main intrinsic issue is computational, since it requires multiple forward models to train.

### 5.5. Conclusions

We detailed three ways to maximize on the expected information gain over a true model of the environment. Each one induces a specific form of surprise. In practice, the expected information gain over a forward model and the learning progress approximate the expected information gain over the true model well. Therefore, it appears that they intuitively and experimentally allow us to explore inherently stochastic environments, but are hard to implement. Models based on prediction error are considerably simpler, but are biased towards inherently stochastic areas. In Table 2, we observe that methods based on prediction error achieve an overall low score on Montezuma’s revenge. Prediction error with *LWM* [65] achieves good performance, presumably because the learned representation is more appropriate. One should be cautious about the low results of the *dynamic-AE* [63] because of the very low number of timesteps. Most methods based on learning progress and information gain over the forward model have not been tested on Montezuma’s revenge, preventing us from directly comparing them to others.

**Table 2 entropy-25-00327-t002:** Comparison between different ways to maximize surprise. Computational cost refers to highly expensive models added to standard RL algorithm. We also report the mean score on *Montezuma’s revenge* (score) and the number of timesteps executed to achieve this score (steps). We gathered results from the original paper and other papers. Our table does not pretend to be an exhaustive comparison of methods, but tries to give an intuition based on their relative advantages.

Method	Computational Cost	Montezuma’s Revenge	
		Score	Steps
Best score [87]	Imitation learning	58,175	200 k
Best IM score [88]	Inverse model, RND	16,800	3500 M
**Prediction error**			
Prediction error with pixels [62]	Forward model	∼160	200 M
Dynamic-AE [63]	Forward model, autoencoder	0	5 M
Prediction error with random features [62]	Forward model, Random encoder	∼250	100 M
Prediction error with VAE features [62]	Forward model, VAE	∼450	100 M
Prediction error with ICM features [62]	Forward model, Inverse model	∼160	100 M
Results from [68]		161	50 M
EMI [68]	Large architecture Error model	387	50 M
Prediction error with LWM [65]	Whitened contrastive loss Forward model	2276	50 M
**Learning progress **			
Learning progress [70,71,72,73,74]	Two forward errors	n/a	n/a
Local learning progress [75,76]	Several Forward models	n/a	n/a
**Information gain over forward model**			
VIME [79]	Bayesian forward model	n/a	n/a
AKL [82]	Stochastic forward model	n/a	n/a
Ensemble with random features [84]	Forward models, random encoder	n/a	n/a
Ensemble with observations [85]	Forward models	n/a	n/a
Emsemble with PlaNet [86]	Forward models, PlaNet	n/a	n/a
JDRX [83]	3 Stochastic Forward models	n/a	n/a

n/a: not available, we did not find the information. RND: Use RND [35]. PlaNet: Use PlaNet [89]. VAE: Use a variational autoencoder [61].

## 6. Novelty Maximization

Novelty quantifies how much a stimuli contrasts with a previous set of experiences [58,90]. More formally, Ref. [58] defends that *an observation is novel when a representation of it is not found in memory, or, more realistically, when it is not “close enough” to any representation found in memory*. Previous experiences may be collected in a bounded memory or distilled in a learned representation.

Several works propose to formalize novelty-seeking as looking for low-density states [91], or, similarly (cf. Section 6.3), states that are different from others [92,93]. In our case, this would result in maximizing the entropy of a state distribution. This distribution can be the t-steps state distribution (cf. Equation (Equation 1)) $H(d_{t}^{\pi }\left(S\right))$ or the entropy of the stationary state-visitation distribution over a horizon *T*
(19)$$\begin{array}{c} H\left(d_{0:T}^{\pi }\left(S\right)\right)=H\left(\frac{1}{T}\sum_{t=1}^{T}d_{t}^{\pi }\left(S\right)\right). \end{array}$$
In practice, these distributions can be approximated with a buffer. This formalization is not perfect and does not fit several intuitions about novelty [58]. Ref. [58] criticizes such a definition by stressing that very distinct and memorable events may have low probabilities of occurring, while not being novel (e.g., a wedding). They suggest that novelty may rather relate to the acquisition of a representation of the incoming sensory data. Following this definition, we propose to formalize novelty-seeking behaviors as those that *actively* maximize the mutual information between states and their representation, I(S;Z)=H(S)−H(S|Z), where *Z* is a low-dimensional space (|Z|≤|S|). This objective is commonly known as the *infomax* principle [94,95,96,97]; in our case, it amounts to actively learning a representation of the environment. Most works focus on actively maximizing the entropy of state distribution while a representation-learning function minimizes H(S|Z). Furthermore, if one assumes that Z=S, the infomax principle collapses to the previously mentioned entropy maximization H(S).

There are several ways to maximize the state-entropy; we separate them based on how they maximize the entropy. We found three kind of methods: expected information gain over a density model, variational inference, and k-nearest neighbors methods.

### 6.1. Information Gain over Density Model

In Section 5, we discussed the maximization of the expected information gain over the true forward model. In practice, one can also maximize the expected information gain over a uniform density model over states $\rho_{U}$. A uniform density model is not given to the agent, but it is possible to make the same approximation as in Section 5.4. We assume the agent tries to learn a density model ρ∈P that approximates the current marginal density distribution of states $p(s^{\prime })$. Then, it can compute the expected information gain over a density model ρ [98]:(20)$$\begin{array}{cc} IG(h,S,A,S^{\prime },P_{U}) & \approx E_{\begin{array}{c} (s,a)∼p(\cdot |h),\\ s^{\prime }∼p\left(\cdot |s,a,h,\phi_{T}\right) \end{array}}D_{KL}(p\left(\rho |h,s^{\prime }\right)||p\left(\rho |h\right)). \end{array}$$
To the best of our knowledge, no work directly optimizes this objective, but it has been shown that this information gain is a lower bound than the squared inverse pseudo-count objective [98], which derives from count-based objectives. In the following, we will review *count* and *pseudo-count* objectives. To efficiently explore its environment, an agent can count the number of times it visits a state and returns to rarely visited states. Such methods are said to be *count-based* [99]. As the agent visits a state, the intrinsic reward associated with this state decreases. It can be formalized with $R\left(s,a,s^{\prime }\right)=\frac{1}{\sqrt{N(s^{\prime })}}$, where N(s) is the number of times that the state *s* has been visited. Although this method is efficient and tractable with a discrete state space, it hardly scales when states are numerous or continuous, since an agent never really returns to the same state. A first solution proposed by [100], called *TRPO-AE-hash*, is to hash the latent space of an auto-encoder fed with states. However, these results are only slightly better than those obtained with a classic exploration policy. Another line of work proposes to adapt counting to high-dimensional state spaces via *pseudo-counts* [98]. Essentially, *pseudo-counts* allow the generalization of the count from a state towards neighborhood states using a learned density model ρ. This is defined as:(21)$$\hat{N}\left(s^{\prime }\right)=\frac{p\left(s^{\prime }|\rho \right)\left(1-p\left(s^{\prime }|\rho^{\prime }\right)\right)}{p\left(s^{\prime }|\rho^{\prime }\right)-p\left(s^{\prime }|\rho \right)}$$
where $\rho^{\prime }\left(s\right)$ computes the density of *s* after having learned on *s*. In fact, ref. [98] show that, under some assumptions, *pseudo-counts* increase linearly with the true counts. In this category, *DDQN-PC* [98] and *DQN-pixelCNN* [101] compute ϕ using, respectively, a context-tree switching model (CTS) [102] and a pixel-CNN density model [103]. Although the algorithms based on density models work on environments with sparse rewards, they add an important complexity layer [101]. One can preserve the quality of observed exploration while decreasing the computational complexity of the pseudo-count by computing it in a learned latent space [104].

There exist several other well-performing tractable exploration methods, such as *RND* [35], *DQN+SR* [105], *RIDE* [106], and *BeBold* [107]. These papers argue that the reward they propose more or less relates to a visitation count estimation.

### 6.2. Variational Inference

Several methods propose to estimate ρ(s) using variational inference [108,109,110,111] based on autoencoder architectures. In this setting, we can use the VAE loss, approximated either as Equation ([Disp-formula FD22b-entropy-25-00327]) [110,112] or Equation ([Disp-formula FD22c-entropy-25-00327]) [111], assuming *z* is a compressed latent variable, p(z) a prior distribution [61], and $q_{decoder}$ a neural network that ends with a diagonal Gaussian.
(22a)$$\begin{array}{cc} \log\rho (s^{\prime }) & \geq E_{\hat{s^{\prime }}∼q_{decoder}\left(\cdot |z\right)}-\log q_{decoder}\left(\hat{s^{\prime }}|z\right)+D_{KL}(q_{encoder}\left(z|s\right)||p\left(z\right)) \end{array}$$
(22b)$$\begin{array}{cc} \; & \approx -\log q_{decoder}\left(s^{\prime }|z\right)+D_{KL}(q_{encoder}\left(z|s^{\prime }\right)||p\left(z\right)) \end{array}$$
(22c)$$\begin{array}{cc} \; & \approx \log\frac{1}{N}\sum_{i=1}^{N}\frac{p(z)}{q_{encoder}\left(z|s^{\prime }\right)}q_{decoder}\left(s^{\prime }|z\right) \end{array}$$
Equation ([Disp-formula FD22c-entropy-25-00327]) is more expensive to compute than Equation ([Disp-formula FD22b-entropy-25-00327]) since it requires decoding several samples, but presumably exhibits less variance. Basically, this estimation allows rewarding an agent [108,110,113] according to: $$R\left(s,a,s^{\prime }\right)=-\log\rho \left(s^{\prime }\right).$$
Ref. [110] maximizes Equation ([Disp-formula FD22c-entropy-25-00327]) by learning new skills that target these novel states (see also Section 7). Using Equation ([Disp-formula FD22b-entropy-25-00327]), ref. [112] approximates Equation ([Disp-formula FD22b-entropy-25-00327]) with the ELBO as used by the VAE. This is similar to *MaxRenyi* [108], which uses the Rény entropy, a more general version of the Shannon entropy, to give more importance to very low-density states. Ref. [109] propose conditioning the state density estimation with policy parameters in order to directly back-propagate the gradient of state-entropy into policy parameters. Although *MaxRenyi* achieves good scores on *Montezuma’s revenge* with pure exploration, maximizing the ground-state entropy may not be adequate, since two closed ground states are not necessarily neighbors in the true environment [114]. Following this observation, *GEM* [115] rather maximizes the entropy of the estimated density of the states considering the dynamic-aware proximity of states, H(Z). However, they do not actively consider H(Z|S).

### 6.3. K-Nearest Neighbors Approximation of Entropy

Several works propose approximating the entropy of a distribution using samples and their k-nearest neighbors [116,117]. In fact, such an objective has already been referred to as a novelty [93]. Assuming $nn_{k}\left(S_{b},s_{i}\right)$ is a function that outputs the k-th closest state to $s_{i}$ in $S_{b}$, this approximation can be written as:(23)$$H\left(S\right)\propto \frac{1}{|S_{b}|}\sum_{s_{i}\in S_{b}}\log||s_{i}-nn_{k}\left(S_{b},s_{i}\right){\left|\right|}_{2}+\chi (|S_{b}\left|\right)+Const$$
where $\chi (S_{b})$ is the digamma function. This approximation assumes the uniformity of states in the ball centered on a sampled state with radius $\left|\right|s_{i}-nn_{k}\left(S_{b},s_{i}\right){\left|\right|}_{2}$ [118], but its full form is unbiased with many samples [116]. Intuitively, it means that the entropy is proportional to the average distance between states and their neighbors. Figure 4 shows how density estimation relates to k-nearest neighbors distance. We clearly see that low-density states tend to be more distant from their nearest neighbors. Few methods [119] provably relate to such estimations, but several approaches take advantage of the distance between state and neighbors to generate intrinsic rewards, making them related to such entropy maximization. For instance, *APT* [120] proposes new intrinsic rewards based on the k-nearest neighbors estimation of entropy:(24)$$\begin{array}{c} R\left(s,a_{t},s^{\prime }\right)=\log(1+\frac{1}{K}\sum_{0}^{K}||f\left(s^{\prime }\right)-nn_{k}\left(f\left(S_{b}\right),f\left(s^{\prime }\right)\right){\left|\right|}_{2}) \end{array}$$
where *f* is a representation function learned with a contrasting loss based on data augmentation [121] and *K* denotes the number of k-nn estimations. By looking for distant state embeddings during an unsupervised pre-training phase, they manage to considerably speed up task-learning in the DeepMind Control Suite. The representation *f* can also derive from a random encoder [122], an inverse model [123], or a contrastive loss that ensures the Euclidean proximity between consecutive states [124,125]. Alternatively, GoCu [126] achieve state-of-the-art results on Montezuma’s revenge by learning a representation with a VAE, and reward the agent based on how distant, in term of timesteps, a state is from a set of k other states.

The contrasting learning loss of the representation can be directly used as a reward [127,128,129]. The agent plays a minimax game. A module learns a representation function with a contrastive loss, which often explicitly estimates the distance of representations with each other [130]. Then, the agent actively challenges the representation by looking for states with a large loss.

Instead of relying on Euclidean distance, one can try to learn a similarity function. *EX${}^{2}$* [131] teaches a discriminator to differentiate states from each other: when the discriminator does not manage to differentiate the current states of those in the buffer, it means that the agent has not visited this state enough, and it will be rewarded. States are sampled from a buffer, implying the necessity to have a large buffer. To avoid this, some methods distill recent states in a prior distribution of latent variables [132,133]. The intrinsic reward for a state is then the KL-divergence between a fixed diagonal Gaussian prior and the posterior of the distribution of latent variables. In this case, common latent states fit the prior, while novel latent states diverge from the prior.

K-nearest neighbors intrinsic rewards have also been employed to improve intra-episode novelty [134]. It contrasts with standard exploration, since the agent looks for novel states in the current episode: typically, it can try to reach all states after every reset. This setting is possible when the policy depends on all its previous interactions, which is often the case when an agent evolves in a POMDP, since the agent has to be able to predict its value function even though varies widely during episodes. This way, ECO [135] and never give up [88] use an episodic memory and learn to reach states that have not been visited during the current episode. Similarly, SDD [136] maximizes the distances between a state and a fixed number of previous states, for example, of close states.

### 6.4. Conclusions

In this section, we reviewed works that maximize novelty to improve exploration with flat policies. We formalized novelty as actively discovering a representation according to the infomax principle, even though most works only maximize the entropy of states/representations of states. Methods based on information gain over density models or variational inference often need a complex neural network to approximate ρ(s). This differs from k-nearest neighbors methods, which only need to compare states with each other and which is often simpler.

In Table 3, we can see that these methods better explore this environment than prediction error-based methods (see Table 2), when using intra-episode novelty mechanisms in particular [88]. This strengthens the hypothesis that novelty maximization better correlates with task rewards than information gain over forward models [59]. While information gain over density models performs well overall in this environment, several methods based on K-nearest neighbors approximations of entropy manage to outperform them. Among the best methods, RND [35] is a simple baseline that achieves important asymptotic performance.

**Table 3 entropy-25-00327-t003:** Comparison between different ways to maximize novelty. Computational cost refers to highly expensive models added to the standard RL algorithm. We also report the mean score on *Montezuma’s revenge* (Score) and the number of timesteps executed to achieve this score (Steps). We gathered results from the original paper and from other papers. Our table does not pretend to be an exhaustive comparison of methods but tries to give an intuition on their relative advantages.

Method	Computational Cost	Montezuma’s Revenge	
		Score	Steps
Best score [87]	Imitation learning	58,175	200 k
Best IM score [88]	Inverse model, RN [35]	16,800	3500 M
**Information gain over density model**			
TRPO-AE-hash [100]	SimHash, AE	75	50 M
DDQN-PC [98]	CTS [102]	3459	100 M ^1^
DQN-PixelCNN [101]	PixelCNN [103]	1671	100 M ^1^
ϕ-EB [104]	Density model	2745	100 M ^1^
DQN+SR [105]	Successor features Forward model	1778	20 M
RND [35]	Random encoder	8152	490 M
[105]		524	100 M ^1^
[68]		377	50 M
RIDE [106]	Forward and inverse models Pseudo-count	n/a	n/a
BeBold [107]	Hash table	∼10,000	2000 M ^1,2^
**Direct entropy maximization**			
MOBE [112], StateEnt [109]	VAE	n/a	n/a
Renyi entropy [108]	VAE, planning	8100	200 M
GEM [115]	Contrastive loss	n/a	n/a
SMM [110]	VAE, Discriminator	n/a	n/a
**K-nearest neighbors entropy**			
EX${}^{2}$ [131]	Discriminator	n/a	n/a
[68]		0	50 M
CB [132]	IB	∼1700	n/a
VSIMR [133]	VAE	n/a	n/a
ECO [135]	Siamese architecture	n/a	n/a
[126]		8032	100 M
DAIM [123]	DDP [137]	n/a	n/a
APT [120]	Contrastive loss	0.2	250 M
SDD [136]	Forward model	n/a	n/a
RE3 [122]	Random encoders ensemble	100	5 M
Proto-RL [124], Novelty [125]	Contrastive loss	n/a	n/a
NGU [88]	Inverse model, RND [35] several policies	16,800	3500 M ^1^
CRL [127], CuRe [128]	Contrastive loss	n/a	n/a
BYOL-explore [129]	BYOL [138]	13,518	3 M
DeepCS [134]	RAM-based grid	3500	160 M
GoCu [126]	VAE, predictor	10,958	100 M

n/a: not available, we did not find the information. IB: Use information bottleneck (IB) method [139]. VAE: Use a variational autoencoder [61]. ^1^ Only provide the number of frames in the paper, we assume they do not use frame skip. ^2^ Result with one single seed.

Some works manage to learn a representation that matches the inherent structure of the environment [124]. This suggests that it is, most of the time, enough to learn a good representation. For instance, refs. [115,124] compute a reward based on a learned representation, but perhaps a bad representation tends to be located in low-density areas. It results that active representation entropy maximization correlates with state-conditional entropy minimization. We are not aware of methods that actively and explicitly maximize I(Z;S) in RL.

## 7. Skill Learning

In our everyday life, nobody has to think about having to move their arms’ muscles to grasp an object. A command to take the object is just issued. This can be performed because an acquired skill can be effortlessly reused.

Skill-learning denotes the ability of an agent to learn a representation of diverse skills. We formalize skill-learning as maximizing the mutual information between the goal g∈G and some of the rest of the contextual states u(τ)∈u(T), denoted as I(G;u(T)), where τ∈T is a trajectory and *u* a function that extracts a subpart of the trajectory (last state, for example). The definition of *u* depends on the wanted semantic meaning of a skill. Let $s_{0}$ refer to the state at which the skill started and *s* a random state from the trajectory, we highlight two settings based on the literature:u(T)=S, the agent learns skills that target a particular state of the environment [140].u(T)=T, the agent learns skills that follow a particular trajectory. This way, two different skills can end in the same state if they cross different areas [141].

Most works maximize I(G;S); unless stated otherwise, we refer to this objective. In the following, we will study the different ways to maximize I(G;S) which can be written under its reversed form I(S;G)=H(G)−H(G|S) or forward form I(G;S)=H(S)−H(S|G) [142]. In particular, we emphasize that:
(25a)$$\begin{array}{cc} -H(G|S) & =\sum_{g\in G,s\in S}p\left(g,s\right)\log p\left(g|s\right) \end{array}$$
(25b)$$\begin{array}{cc} \; & =E_{\begin{array}{c} g∼p(g)\\ s∼\pi^{g} \end{array}}\log p\left(g|s\right) \end{array}$$
where, to simplify, p(g) is the current distribution of goals (approximated with a buffer) and $s∼\pi^{g}$ denotes the distribution of states that result from the policy that achieves *g*. Note that p(g,s)=p(s|g)p(g).

In this section, we first focus on methods that assume they can learn all skills induced by a given goal space/goal distribution, and which assign parts of trajectories to every goal. The second set of methods directly derives the goal space from visited states, so that there are two different challenges that we treat separately: the agent has to learn to reach a selected goal, and it must maximize the diversity of goals it learns to reach. We make this choice of decomposition because some contributions focus on only one part of the objective function.

### 7.1. Fixing the Goal Distribution

The first approach assumes the goal space is arbitrarily provided, except for the semantic meaning of a goal. In this setting, the agent samples goals uniformly from *G*, ensuring that H(G) is maximal, and it progressively assigns all possible goals to a part of the state space. To do this assignment, the agent maximizes the reward provided by Equation ([Disp-formula FD25b-entropy-25-00327])
(26)$$R\left(g,s,a,s^{\prime }\right)=-\log q_{\omega }\left(g|s^{\prime }\right)$$
where $q_{\omega }\left(g|s^{\prime }\right)$ represents a learned discriminator (often a neural network) that approximates $p(g|s^{\prime })$.

At first, we focus on a discrete number of skills, where p(g) represents a uniform categorical distribution. Figure 5 sums up the learning process with two discrete skills. 1: Skills and discriminator $q_{\omega }$ are randomly initialized; 2: the discriminator tries to differentiate the skills with states *s* from its trajectories, in order to approximate p(g|s); 3: skills are rewarded with Equation (Equation 26) in order to make them go in the area assigned to it by the discriminator; 4: finally, skills are clearly distinguishable and target different parts of the state space. *SNN4HRL* [143] and *DIAYN* [140] implement this procedure by approximating *g* with, respectively, a partition-based normalized count and a neural network. *VALOR* [144] also uses a neural network, but discriminates discrete trajectories. In this setting, the agent executes one skill per episode.

*HIDIO* [146] sequentially executes skills, yet it is not clear how they manage to avoid forgetting previously learned skills. Maximizing $I(G;S|S_{0})$ like *VIC* [147] or $I(G;S_{0}|S)$ with *R-VIC* [148] makes it hard to use a uniform (for instance) $H(G|S_{0})$, because every skill may not be executable everywhere in the state space. Therefore, they also maximize the entropy term with another reward bonus, similar to $\log p(g|s_{0})$. They learn discriminant skills, but still struggle to combine them on complex benchmarks [148]. Keeping p(g) uniform, *DADS* [149] maximizes the forward form of mutual information $I\left(S;G|S_{0}\right)=H\left(S|S_{0}\right)-H\left(S|G,S_{0}\right)$ by approximating $p(s|s_{0})$ and $p(s|s_{0},g)$. This method makes possible to plan over skills, and can combine several locomotion skills. However, this requires several conditional probability density estimations on the ground state space, which may scale badly to higher-dimensional environments.

These methods tend to stay close to their starting point [142] and do not learn skills that cover the whole state space. In fact, it is easier for the discriminator to overfit over a small area than to make a policy go to a novel area, this results in a lot of policies that target a restricted part of the state space [150]. Accessing the whole set of true possible states and deriving the set of goals by encoding states can considerably improve the coverage of skills [142].

Heterogeneous methods address the problem of overfitting of the discriminator. The naive way can be to regularize the learning process of the discriminator. *ELSIM* [42] takes advantages of L2 regularization and progressively expands the goal space *G* to cover larger areas of the state space, and [150] proposes to use spectral normalization [151]. More consistent dynamic-aware methods may further improve regularization; however, it remains hard to scale the methods to many skills which are necessary to scale to a large environment. In the above-mentioned methods, the number of skills greatly increases [42,144] and the discrete skill embedding does not provide information about the proximity of skills. Therefore, learning a continuous embedding may be more efficient.

The prior uniform distribution p(g) is far more difficult to set in a continuous embedding. One can introduce the *continuous DIAYN* [146,150] with a prior $p\left(G\right)=N\left(0^{d},I\right)$, where *d* is the number of dimensions, or the *continuous DADS* with a uniform distribution over [−1;1] [149], yet it remains unclear how the skills could adapt to complex environments, where the prior does not globally fit the inherent structure of the environment (e.g., a disc-shaped environment). *VISR* [152] seems to, at least partially, overcome this issue with a long unsupervised training phase and successor features. They uniformly sample goals on the unit-sphere and compute the reward as a dot product between unit-normed goal vectors and successor features $\log q_{\omega }\left(g|s\right)=\phi_{successor}{\left(s\right)}^{T}g$.

### 7.2. Achieving a State Goal

In this section, we review how current methods maximize the goal-achievement part of the objective of the agent, $-H(S_{g}|S)$, where $S_{g}$ refers to the goal-relative embedding of states. We temporally set aside $H(S_{g})$ and we will come back to this in the next subsection, Section 7.3, mainly because the two issues are tackled separately in the literature. Obviously, maximizing $-H(S_{g}|S)$ can be written as:(27)$$\begin{array}{cc} -H(S_{g}|S) & =\sum_{S_{g},S}p\left(s_{g},s\right)\log p\left(s_{g}|s\right)=E_{\begin{array}{c} s_{g}∼p\left(s\right)\\ s∼\pi^{g} \end{array}}\log p\left(s_{g}|s\right) \end{array}$$
where, to simplify, p(s) is the current distribution of states (approximated with a buffer) and $s∼\pi^{g}$ denotes the distribution of states that results from the policy that achieves *g*. If $\log p(s_{g}|s^{\prime })$ is modeled as an unparameterized Gaussian with a unit-diagonal co-variance matrix, we have $\log p\left(s_{g}|s^{\prime }\right)\propto -\left|\right|s_{g}-s^{\prime }{\left|\right|}_{2}^{2}+Const$ so that we can reward an agent according to:(28)$$R\left(s_{g},s,a,s^{\prime }\right)=-\left|\right|s_{g}-s^{\prime }{\left|\right|}_{2}^{2}.$$
This means that if the goal is a state, the agent must minimize the distance between its state and the goal state. To achieve this, it can take advantage of a goal-conditioned policy $\pi^{s_{g}}\left(s\right)$.

This way, *hierarchical actor–critic (HAC)* [153] directly uses the state space as a goal space to learn three levels of options (the options from the second level are selected to fulfill the chosen option from the third level). A reward is given when the distance between states and goals (the same distance as in Equation (Equation 28)) is below a threshold. Similar reward functions can be found in [154,155]. Related to these works, *HIRO* [156] uses as a goal the difference between the initial state and the state at the end of the option $f\left(T\right)=S_{f}-S_{0}$.

This approach is relatively simple, and does not require extra neural networks. However, there are two problems with using the state space in the reward function. Firstly, a distance (such as L2) makes little sense in a very large space, such as images composed of pixels. Secondly, it is difficult to make a manager’s policy learn on a too-large action space. Typically, an algorithm with images as goals can imply an action space of 84×84×3 dimensions for a goal-selection policy (in the case of an image with standard shape). Such a wide space is currently intractable, so these algorithms can only work on low-dimensional state spaces.

To tackle this issue, an agent can learn the low-dimensional embedding of space $\phi_{e}$ and maximize the reward of Equation (Equation 29) using a goal-conditioned policy $\pi^{f(s_{g})}\left(s\right)$:(29)$$R\left(s_{g},s,a,s^{\prime }\right)=-||f\left(s_{g}\right)-f\left(s^{\prime }\right){\left|\right|}_{2}^{2}.$$

Similarly to Equation (Equation 28), this amounts to maximize $-H(f\left(S_{g}\right)|f\left(S\right))$. *RIG* [157] proposes to build the feature space independently with a variational auto-encoder (VAE); however, this approach can be very sensitive to distractors (i.e., useless features for the task or goal, inside states) and does not allow correctly weighting features. Similar approaches also encode part of trajectories [141,158] for similar mutual information objectives. *SFA-GWR-HRL* [159] uses unsupervised methods such as the algorithms of *slow features analysis* [160] and *growing when required* [161] to build a topological map. A hierarchical agent then uses nodes of the map, representing positions in the world, as a goal space. However, the authors do not compare their contribution to previous approaches.

Other approaches learn a state-embedding that captures the proximity of states with contrasting losses. For instance, *DISCERN* [162] learns the representation function by maximizing the mutual information between the last state representation and the state goal representation. Similarly to works in Section 7.1, the fluctuations around the objective allow bringing states around $s_{g}$ closer to it in the representation. *LEXA* [163] learns to measure the similarity between two states by training a neural network to predict the number of steps that separate two states. Then, the similarity between a goal and the states can reward the goal-conditioned policy. More explicitly, the representation of *NOR* [164] maximizes $I(f\left(S_{t+k}\right);f\left(S_{t}\right),A_{t:t+k})$ and the one of *LESSON* [165] maximizes $I(f\left(S_{t+1}\right);f\left(S_{t}\right))$; *LESSON* and *NOR* target a change in the representation and manage to navigate in a high-dimensional maze while learning the intrinsic Euclidean structure of the mazes (cf. Table 4). Their skills can be reused in several environments with similar state spaces. However, experiments are made in two-dimensional embedding spaces, and it remains unclear how relevant goals may be as states change in an embedding space with higher dimensions.The more the number of dimensions increases, the more difficult it will be to distinguish possible skills from impossible skills in a state. In addition, *LESSON* and *NOR* need dense extrinsic rewards to learn to select the skills to execute. Thus, they generate tasks with binary rewards at a location uniformly distributed in the environment, such that the agent learns to achieve the tasks from the simplest to the hardest. This progressive learning generates a curriculum, helping to achieve the hardest task.

### 7.3. Proposing Diverse State Goals

To make sure the agent maximizes the mutual information between its goals and all visited states, it must sample a diverse set of goal-states. In other words, it has to maximize $H(S_{g})$ but through goal selection, rather than with an intrinsic bonus as in Section 6. Similarly to works on novelty (cf. Section 6), such entropy maximization along with skill acquisition (cf. Section 7.2) tackles the exploration challenge, but without facing catastrophic forgetting (cf. Section 8.1) since the agent does not forget its skills.

A naive approach would be to generate random values in the goal space, but this brings a considerable problem: the set of achievable goals is often a very small subset of the entire goal space. To tackle this, a first approach can be to explicitly learn to differentiate these two sets of goals [166,167], using, for example, generative adversarial networks (GAN) [166,168], which is ineffective in complex environments [111]. Other works obtain good results on imagining new goals, but using a compoundable goal space, given [169] or learned with a dataset, see [170]; results show it may be a strong candidate for object-based representations. In contrast, in a more general case, an agent can simply set a previously met state as a goal; in this way, it ensures that goals are reachable, since they have already been achieved. In the rest of this section, we focus on this set of methods.

In *RIG* [157], the agent randomly samples states as goals from its buffer, but it does not increase the diversity of states, and thus, the diversity of learned skills. [111] showed, theoretically and empirically, that, by sampling goals following a more uniform distribution over the support of visited states than the “achieved” distribution, the distribution of states of the agent can converge to the uniform distribution. Intuitively, the agent just samples low-density goals more often, as illustrated in Figure 6. There are several ways to increase the importance of low-density goal-states that we introduce in the following.

*DISCERN* [162] proposes to sample the support of visited states uniformly with a simple procedure. Every time the agent wants to add an observation to its buffer, it randomly samples another observation from its buffer and only keeps the one that is the furthest from all other states of the buffer. This way, it progressively builds a uniform distribution of states inside its buffer. However, it uses the Euclidean distance to compare images, which may not be relevant. Other approaches select the state that has the lower density (*OMEGA*) [154] according to a kernel density estimation or use the rank of state-densities [171] estimated with a variational Gaussian mixture model [172]. In contrast with them, *skew-fit* [111] provides more flexibility on how uniform one wants its distribution of states. *Skew-fit* extends RIG and learns a parameterized generative model $q_{\rho }\left(S\right)\approx p\left(S\right)$ and skews the generative model (VAE) with the ratio:(30)$$q_{\rho }{\left(s\right)}^{\alpha_{skew}}$$
where $\alpha_{skew}<0$ determines the speed of uniformization. This way it gives more importance to low-density states. The, n it weights all visited states according to the density approximated by the generative model at the beginning of each epoch, which is made of a predefined number of timesteps. Skew-fit manages to explore image-based environments very efficiently. As highlighted in [114], this ratio applied to a discrete number of skills amounts to reward a Boltzmann goal-selection policy with
(31)$$R\left(s_{g}\right)=\left(1+\alpha_{skew}\right)\log p\left(s_{g}\right).$$
With a different objective, *GRIMGREP* [173] partitions the VAE embedding of skew-fit with a Gaussian mixture model [174] to estimate the learning progress of each partition and avoid distractors. The density weighting can also operate in a learned embedding. *HESS* [175] partitions the embedding space of *LESSON* and rewards it with a variant of a count-based bonus (see Section 5). It improves exploration in a two-dimensional latent embedding, but the size of partitions may not scale well if the agent considers more latent dimensions. In contrast, *DisTop* [114] dynamically clusters a dynamic-aware embedding space using a variant of growing when required [161]; they estimate the density of the state according to how much its partition contains states, and skew the distribution of samples similarly to skew-fit. *HESS* and *DisTop* demonstrate their ability to explore and navigate with an ant inside a complex maze without extrinsic rewards. As shown in [114] (illustration in Figure 6c), it is also possible to use extrinsic rewards to weight the distribution of sampled state goals.

### 7.4. Conclusions

We found two main ways to discover skills. The first one provides a goal space and assigns goals to areas of the state space. There are empirical evidences emphasizing that an actor struggles to learn and sequentially execute skills that target different areas of the state space. The second method derives the goal space from the state space with a representation learning method, and over-weights the sampling of low-density visited areas. This set of works showed the ability to hierarchically navigate in simple environments using moderately morphologically complex agents.

In Table 4, we can make two observations. 1: methods that do not propose diverse goal-states require an implicit curriculum to learn the ant maze task [156,165] (*curriculum* column); 2: contrasting representations seem crucial to avoid using a hand-defined goal space such as the (x, y) coordinated (*goal space* column) [164,175]. For methods in “fixing the goal distribution”, we did not find a representative and widely used evaluation protocol/environment among the published works. However, as an example, several qualitative analyses emphasize the diversity of behaviors that can be learned by an *ant* [140,149].

**Table 4 entropy-25-00327-t004:** Summary of papers regarding learning skills through mutual information maximization. We selected the ant maze environment to compare methods, since this is the most commonly used environment. We did not find a common test setting allowing for a fair comparison of methods in “Fixing the goal distribution”. The scale refers to the size of the used maze. Goal space refers to the a priori state space used to compute goals, from the less complex to the more complex: (x, y); 75-dimensional top-view of the maze (T-V); top-view + proprioceptive state (T-V + P). Curriculum refers to whether the authors use different goal locations during training, creating an implicit curriculum that makes learning to reach distant goals from the starting position easier. The score, unless stated otherwise, refers to the success rate in reaching the furthest goal.

Fixing the goal distribution					
SNN4HRL [143], DIAYN [140], VALOR [144]					
HIDIO [146], R-VIC [148], VIC [147]					
DADS [149], continuous DIAYN [150], ELSIM [42]					
VISR [152]					
Methods	Scale	Goal space	Curriculum	Score	Steps
**Achieving a state goal**					
HAC [153]	n/a	n/a	n/a	n/a	n/a
HIRO [156]	4x	(x, y)	Yes	∼0.8	4 M
RIG [157]	n/a	n/a	n/a	n/a	n/a
SeCTAR [141]	n/a	n/a	n/a	n/a	n/a
IBOL [158]	n/a	n/a	n/a	n/a	n/a
SFA-GWR-HRL [159]	n/a	n/a	n/a	n/a	n/a
NOR [164]	4x	T-V + P	Yes	∼0.7	10 M
[165]	4x	T-V + P	Yes	∼0.4	4 M
LESSON [165]	4x	T-V + P	Yes	∼0.6	4 M
**Proposing diverse state goals**					
Goal GAN [166]	1x	(x, y)	No	Coverage 0.71%	n/a
[111]	n/a ($\tilde{1}$6x)	(x, y)	No	All goals distance∼7	7 M
FVC [167]	n/a	n/a	n/a	n/a	n/a
Skew-fit [111]	n/a ( $\tilde{1}$6x)	(x, y)	No	All goals distance∼1.5	7 M
DisTop [114]	4x	T-V	No	∼1	2 M
DISCERN [162]	n/a	n/a	n/a	n/a	n/a
OMEGA [154]	4x	(x, y)	No	∼1	4.5 M
CDP [171]	n/a	n/a	n/a	n/a	n/a
[111]	n/a ( $\tilde{1}$6x)	(x, y)	No	All goals distance∼7.5	7 M
GRIMGREP [173]	n/a	n/a	n/a	n/a	n/a
HESS [175]	4x	T-V + P	No	success rate∼1	6 M

## 8. Outlooks of the Domain

In this section, we take a step back and thoroughly analyze the results of our overall review. We first study the exploration process of flat intrinsic motivation in comparison with hierarchical intrinsic motivations in Section 8.1; this will motivate our focus on the challenges induced by learning a deep hierarchy of skills in Section 8.2. Finally, in Section 8.3, we discuss how flat and hierarchical intrinsic motivations could and should cohabit in such a hierarchy.

### 8.1. Long-Term Exploration, Detachment and Derailment

The most challenging used benchmark in flat intrinsic motivations (surprise and novelty) is *Montezuma’s revenge*, yet very sparse reward games such as *Pitfall!* are not currently addressed and should be investigated. In *Pitfall!*, the first reward is reached only after multiple rooms, and the game requires specific action sequences to go through each room. State-of-the-art IM methods [101] complete the game with 0 mean rewards. Contrastingly, imitation RL methods [87,176] are insensitive to such a specific reward, and thus, exceed IM methods with a mean reward of 37,232 on *Montezuma’s revenge* and 54,912 on *Pitfall!*. Even though these methods use expert knowledge, this performance gap exhibits their resilience to long-term rewards. This shows that flat IMs are still far from solving the overall problem of exploration.

Furthermore, we want to emphasize that the challenge is harder when the intrinsic reward itself is sparse [35]. *Montezuma’s revenge* it is about avoiding using a key too quickly, in order to be able to use it later. In fact, it looks like there is an exploration issue in the intrinsic reward function. Intrinsic reward can guide the exploration with the condition that the agent finds this intrinsic reward. There may be two reasons causing the intrinsic reward to be sparse:1.The first comes from partial observability, with which most models are incompatible. Typically, if an agent has to push a button and can only see the effect of this pushing after a long sequence of actions, density models and predictive models may not provide meaningful intrinsic rewards. There would be too large a distance between the event “pushing a button” and the intrinsic reward.2.Figure 7 illustrates the second issue, called *detachment* [177,178]. This results from a distant intrinsic reward coupled with catastrophic forgetting. Simply stated, the RL agent can forget the presence of an intrinsic reward in a distant area: it is hard to maintain the correct q-value that derives from a distant, currently unvisited, rewarding area. This is emphasized in on-policy settings.

Pursuing such distant intrinsic rewards may be even harder due to the possible *derailment* issue [177,178]. Essentially, an agent may struggle to execute a long sequence of specific actions needed to reach a distant rewarding area, because the local stochasticity incites local dithering along the sequence. Detachment motivates the need for a hierarchical exploration [178] and derailment motivates frontier-based exploration [179], which consists of deterministically reaching the area to explore before starting exploration.

### 8.2. Deeper Hierarchy of Skills

Most works presented in Section 7 abstract actions on a restricted number of hierarchies (generally one hierarchy). This is necessary for understanding the mechanism of abstraction, but we want to argue that imposing deeper hierarchies could considerably enhance the semantic comprehension of the environment of an agent. Organisms are often assumed to deal with the composition of behaviors, which, in turn, serve as building blocks for more complex behaviors [180]. This way, using a limited vocabulary of skills makes it easier to avoid the curse of dimensionality associated with the redundancy of a whole set of ground behaviors.

Our surveyed works [65,114,115,164,165] already propose to learn the representations using the slowness principle [160] which assumes temporally close states should be similarly represented. By configuring the time-extension of the representation, one may focus on different semantic parts of the state space. This can be seen in Section 3.2; 1: the agent can learn a very low level representation that provides skills that can manipulate torques of a creature [114]; 2: skills can also orientate an agent in a maze by extracting (x, y) coordinates from a complex state representation [175]. While they do not try to combine and learn several representations at the same time, further works could consider separating different parts of states (e.g., agent positions and object positions [181]) or learning these representations at different time scales.

In a developmental process, multi-level hierarchical RL questions the ability of the agent to learn all policies of the hierarchy simultaneously. This obviously relates to the ability of organisms to continually learn throughout their lifetime, but in a more practical way, it may allow focusing on the learning process of skills that are interesting for higher-level skills. This focus avoids learning everything in the environment [114], which is hard and obviously not accomplished by biological organisms. For instance, most people cannot do a somersault.

Considering a goal representation that changes over time introduces new issues for the agent. In this case, the goal-conditioned policy may be perturbed by the changes of inputs and may no longer be able to reach the goal [175]. Current methods consider 1: developmental periods (unsupervised pre-training [182]); 2: to modify the representation every k-steps epoch [111]; 3: to impose changes of the representation slowly [175]. Further works may thoroughly investigate the relation and transitions between these methods, since they can relate to the concept of critical periods [183,184]. Critical periods assume that the brain is more plastic at some periods of development in order to acquire specific knowledge. Despite this mechanism, the brain keeps learning slowly throughout the lifetime. In the hierarchy of skills, the introduction of a new level may first result in a quick/plastic learning process, followed by slower changes. Furthermore, the level of skill acquisition may impact the bounds of critical periods [185,186], allowing introductions of new levels at the right time.

### 8.3. The Role of Flat Intrinsic Motivations

In Section 8.1, we essentially criticized the limited role that flat intrinsic motivations such as surprise or novelty can play in favor of exploration, and we hypothesized in Section 8.2 that deeper hierarchies could make emerge an understanding of more complex affordances. Then, what are the roles of surprise and novelty? We saw in Section 6 that novelty-seeking behaviors allow learning a correct representation of the whole environment; this can be a basis for learning diverse skills. While some methods consider a goal as a state and manage to avoid using novelty bonuses [111], this is harder to do when skills have a different semantic (such as a change in the state space). Ref. [164] provide a meaningful example of this: the agent acts to simultaneously discover a representation of the environment and achieve upper-level goals.We leave aside the interest of surprise for learning a forward model that could be used for planning [89] and rather focus on the learning process. Surprise amounts to looking for the learning progress of forward models so that, in a hierarchy of skills, it quantifies whether skills can currently be better learned or not. This links surprise to curriculum learning [187], i.e., can we find a natural order to efficiently learn skills? For example, assuming an agent wants to learn to reach a state goal in a maze, it would be smarter to learn to start learning skills that target goals close to a starting position, and to progressively extend its goal selection while learning other skills. Several strategies have been proposed to smartly hierarchically select goals [16,169]. Among them, the competence progress metric, which measures whether a skill is currently being improved, seems highly efficient [41,169,188,189]. We think that competence progress and the learning progress of a forward model over skills are two sides of the same coin. In practice, when trying to predict the temporally extended consequences of a skill (rather than the one of a ground action), the success of the prediction highly correlates with whether the skill reliably reaches its goal, i.e., it is mastered. Similarly, a temporally-extended prediction can improve only if the associated skill improves, assuming an uncontrolled skill does not always go in the same wrong state.

To sum up, we propose that the role of surprise and novelty may actually be to support the learning of skills. Novelty-seeking helps to learn the representation required by the skill-learning module, and surprise speeds up the maximization of the skill-learning objective. They may interact as a loop: first, the agent learns a new representation, then it evaluates surprise to select which skill to improve and the skill-learning process starts. Considering this, it would result in several surprises and novelties: an agent can experiment a novel or surprise interaction for a level of decision, yet it does not mean other levels would be surprised. This emphasizes the multidimensionality and relativity of the notions of surprise and novelty [190]; only a part of the incoming stimuli may arouse the agent.

## 9. Conclusions

In this survey, we have presented the current challenges faced by DRL, namely, 1: learning with *sparse rewards* through exploration; 2: *building a hierarchy of skills* in order to make easier credit assignment, exploration with *sparse rewards*, and *transfer learning*.

We identified several types of IM to tackle these issues which we classified into three categories based on a maximized information theoretic objective, which are *surprise*, *novelty*, and *skill-learning*. Surprise- and novelty-based intrinsic motivations implicitly improve flat exploration, while skill-learning allows the creation of a hierarchy of reusable skills that also improves exploration.

**Surprise** results from maximizing the mutual information between the true model parameters and the next state of a transition, knowing its previous state, its previous action, and the history of interactions. We have shown that it can be maximized through three sets of works: prediction error, learning progress, and information gain over forward models. In practice, we found that the determinism assumption underpinning prediction error methods complicates their application. Good approximations of surprise are notably useful to allow exploration in stochastic environments. The next challenges may be to make good approximations of surprise tractable.

**Novelty**-seeking can be assimilated to learning a representation of the environment through the maximization of mutual information between states and their representation. The most important term to actively maximize appears to be the entropy of state or representation, which can be approximated in three ways: 1: one can reward the expected information gain over a density model; 2: one can reward according to the parametric density of its next state, but it is complicated to estimate; 3: one can also reward an agent according to the distance between a state and already-visited states, making the approach tractable, in particular, when the agent learns a dynamic-aware representation. We found these methods to achieve state-of-the-art performance on the hard exploration task of Montezuma’s revenge. We expect future works to benefit from directly looking for good representations rather than the uniformity of states.

Finally, using a **skill-learning** objective that amounts to maximizing the mutual information between a goal and a set of trajectories of the corresponding skill, an agent can learn hierarchies of temporally extended skills. Skills can be directly learned by attributing part of a fixed goal space to areas, but it remains to be seen how well goals can be embedded in a continuous way, and whether approaches may be robust when skills are sequentially executed. The second approach derives the goals space from the state space, often through a time-contrasting loss, and expands the skill set by targeting low-density areas. These methods manage to explore an environment while being able to return to previously visited areas. It remains to be demonstrated how one could create larger hierarchies of skills.

The three objectives are compatible, and we have discussed how they could interact to provide a robust exploration with respect to the *detachment* issue, along with reusable hierarchical skills, a quick and focused skill acquisition, and multi-semantic representations.

## Figures and Tables

**Figure 1 entropy-25-00327-f001:**
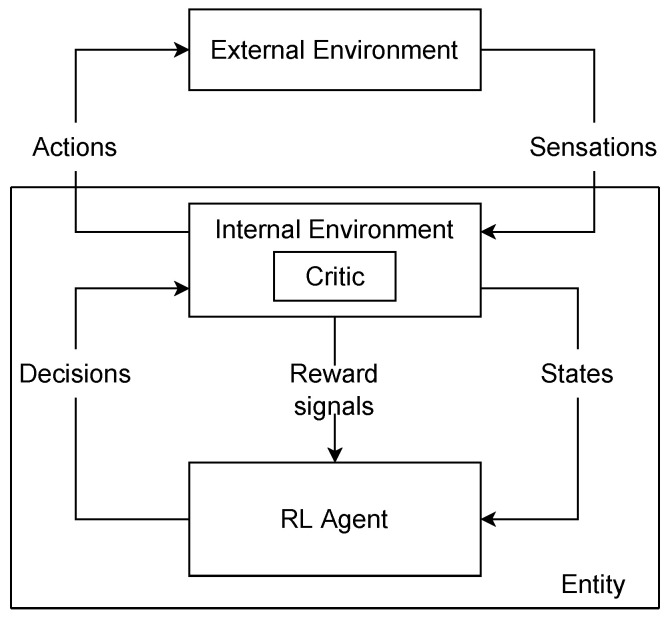
Model of RL integrating IM, taken from [29]. The environment is factored into an internal and external environment, with all rewards coming from the former.

**Figure 2 entropy-25-00327-f002:**
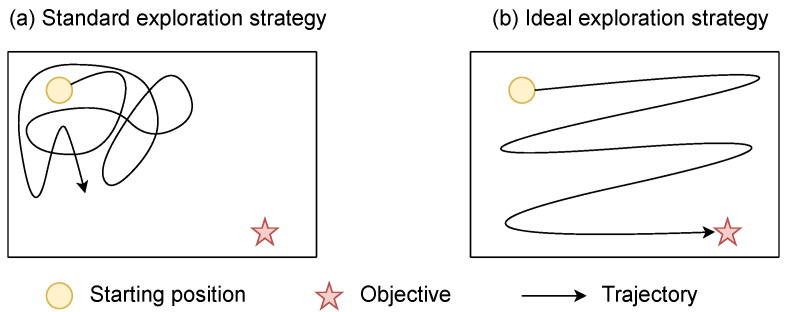
Example of a very simple sparse reward environment, explored by two different strategies. The agent, represented by a circle, strives to reach the star. The reward function is one when the agent reaches the star, and zero otherwise. (**a**) The agent explores with standard methods such as ϵ-greedy; as a result, it stays in its surrounded area because of the temporal inconsistency of its behavior. (**b**) We imagine an ideal exploration strategy where the agent covers the whole state space to discover where rewards are located. The fundamental difference between the two policies is the volume of the state space explored for a given time.

**Figure 3 entropy-25-00327-f003:**
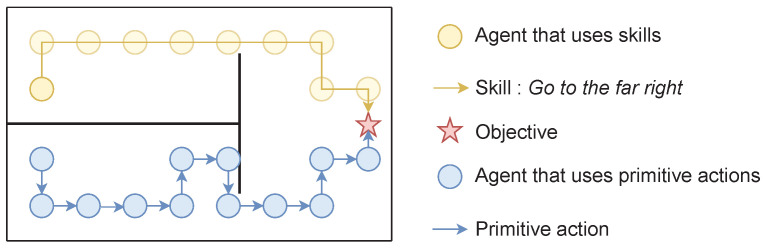
Example of two policies in a simple environment, one uses *skills* (yellow), the other one only uses primitive actions (blue). Agents have to reach the star.

**Figure 4 entropy-25-00327-f004:**
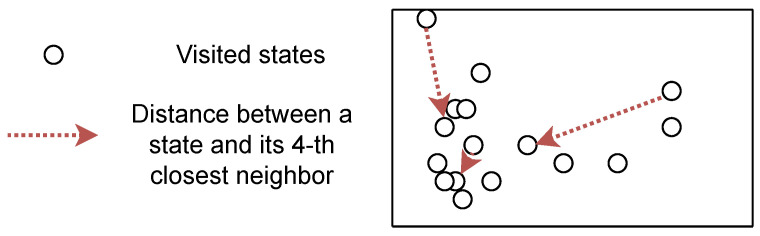
Illustration of the correlation between density and the fourth-nearest neighbor distance.

**Figure 5 entropy-25-00327-f005:**
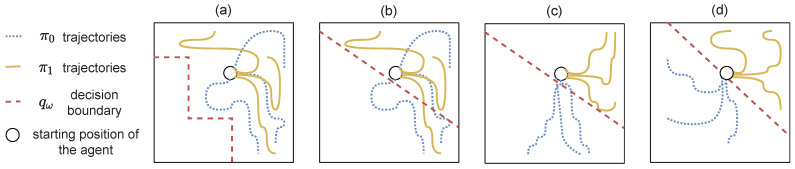
Illustration of the implicit learning steps of algorithms that use a fixed goal distribution. (**a**) Skills are not learnt yet. The discriminator randomly assigns partitions of the state space to goals. (**b**) The discriminator tries unsuccessfully to distinguish the skills. (**c**) Each skill learns to go in the area assigned to it by the discriminator. (**d**) Skills locally spread out by maximizing action entropy [145]. The discriminator successfully partitions the areas visited by each skill.

**Figure 6 entropy-25-00327-f006:**
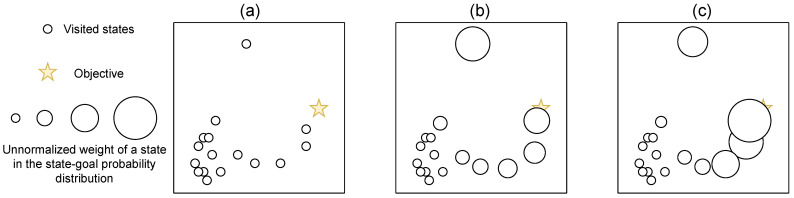
Illustration of the re-weighting process. (**a**) Probability of visited states to be selected as goals before re-weighting; (**b**) Probability of visited states to be selected as goals after density re-weighting; (**c**) Probability of visited states to be selected as goals after density/reward re-weighting. This figure completes and simplifies the figure from [111].

**Figure 7 entropy-25-00327-f007:**
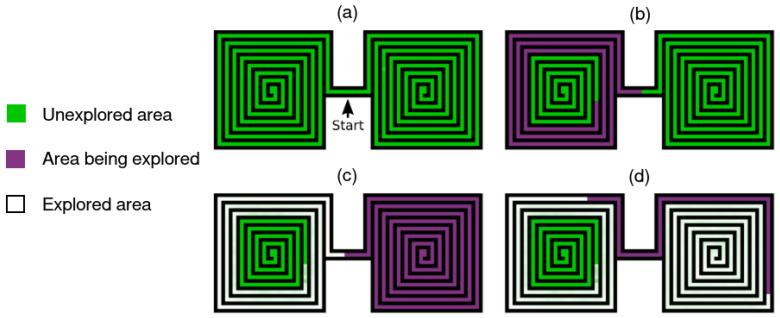
Illustration of the *detachment* issue. Image extracted from [177]. Green color represents intrinsically rewarding areas, white color represents no-reward areas, and purple areas are currently being explored. (**a**) The agent has not explored the environment yet. (**b**) It discovers the rewarding area at the left of its starting position and explores it. (**c**) It consumes close intrinsic rewards on the left side, thus it prefers gathering the right-side intrinsic rewards. (**d**) Due to catastrophic forgetting, it forgets how to reach the intrinsically rewarding area on the left.

**Table 1 entropy-25-00327-t001:** Summary of our taxonomy of intrinsic motivations in DRL. The function *u* outputs a part of the trajectories T. *Z* and *G* are internal random variables denoting state representations and self-assigned goals, respectively. Please refer to the corresponding sections for more details about methods and notations. The reward function aims to represent the one used in the category.

**Surprise**: $I(S^{\prime };\Phi_{T}|h,S,A)$, Section 5			
Formalism	Prediction error	Learning progress	Information gain over forward model over forward model
Sections	Section 5.3	Section 5.2	Section 5.4
Rewards	$\left|\right|s^{\prime }-{\hat{s}}^{\prime }{\left|\right|}_{2}^{2}$	$\Delta ||s^{\prime }-{\hat{s}}^{\prime }{\left|\right|}_{2}^{2}$	$D_{KL}(p\left(\Phi |h,s,a,s^{\prime }\right)||p\left(\Phi |h\right))$
Advantage	Simplicity	**Stochasticity robustness **	**Stochasticity robustness**
**Novelty**: I(S;Z), Section 6			
Formalism	Information gain over density model	Parametric density	K-nearest neighbors
Sections	Section 6.1	Section 6.2	Section 6.3
Rewards	$\frac{1}{\sqrt{\hat{N}\left(s^{\prime }\right)}}$	$-\log\rho (s^{\prime })$	$\log(1+\frac{1}{K}\sum_{0}^{K}||f\left(s^{\prime }\right)-nn_{k}\left(f\left(S_{b}\right),f\left(s^{\prime }\right)\right)|{|}_{2})$
Advantage	Good exploration	Good exploration	**Best exploration**
**Skill-learning**: I(G;u(T)), Section 7			
Formalism	Fixed goal distribution	Goal-state achievement	Proposing diverse goals
Sections	Section 7.1	Section 7.2	Section 7.3
Rewards	$\log p(g|s^{\prime })$	$-\left|\right|s_{g}-s^{\prime }{\left|\right|}_{2}^{2}$	$\left(1+\alpha_{skew}\right)\log p\left(s_{g}\right)$
Advantage	Simple goal sampling	**High-granularity skills**	**More diverse skills**

## Data Availability

Data sharing not applicable.

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
