# Peer review of "An Information-Theoretic Perspective on Intrinsic Motivation in Reinforcement Learning: A Survey"

_entropy, 2023, doi:10.3390/e25020327_

Round 1

Reviewer 1 Report

Please find my comments in the attached file. 

Author Response

Thank you for your in-depth review. We highlighted in red our modifications to the manuscript.

1) Related works

Dreamer papers: To the best of our knowledge, Hafner et al. (2019) and Hafner et al. (2020a) do not incorporate intrinsic motivations. Hafner et al. (2020a) do not propose a new model of intrinsic motivation. Its objective, different from ours, aims to show how a wide set of approaches (beyond intrinsic motivation) relates to the minimization of a KL divergence. However we cite (Sekar et al (2020)) which adds a disagreement-based IM on top of dreamer and we added (Mendonca et al) which learns skills with Dreamer.

Information-theoretic objectives: We added a small paragraph in the introduction to clarify the scope of the survey with respect to some other bodies of works, including works you suggest. A myriad of other information theoretic objectives exist, but this is not a survey on information-theoretic IMs. The objective of this paper is rather to show the relation of these 3 objectives with a large body of works that tackle the issues of explorations and skill learning, and analyze this part of the literature thanks to the introduced taxonomy.

2) Comparison of techniques

The comparison indeed has limitations as we can not make a complete benchmarking of approaches on several environments. We modified the tables and our conclusions in several ways, following your first suggestion:
- We decided to remove the Stochasticity column from the tables. We think a binary Yes/No indeed does not reflect well enough the contributions of the papers. In addition the data was too sparse. We leave this analysis to the main text.
- We made sure to not overstate our conclusions by extrapolating the results on Montezuma's Revenge to other environments.
- We made sure to not overstate on the missing performances.
- We also further argued for the relevance of Montezuma's revenge for this comparison: 1- A simple DQN achieves a 0 score because of exploration unlike most of IMs methods; 2- it is very often used for comparison in papers tackling exploration: 3- the performances increases as the agent explores new rooms, making the score related to exploration. We also think that the correlation between task rewards and human behaviors in Montezuma's Revenge further validates this choice (Matusch et al. (2020)). We moved these arguments to Section 4 when we first mention the tables to avoid a redundant explanation in the conclusion of surprise and novelty sections.

While the main interest of the tables is also to sum up all methods with their associated paper, we assume we can draw some conclusions from the Montezuma's scores:
- Prediction error are bad at exploring Montezuma's revenge, in comparison to novelty methods.
- K-NN methods overall outperform methods based on information gain over density model in Montezuma's revenge.
As explained in the paper, it mostly aims to give a clue on how each method compare to the others.

3) Compact sets.

Thank you for the detailed comment. We fixed it by assuming X has a compact domain.
We decided to keep the rest of our definition this way since we often refer to entropy of continuous variables throughout the manuscript and to the importance to make its probability density distribution closer to a uniform distribution, in particular in Section 7. In each environment, the value of the entropy of the uniform distribution should not impact our analysis as long as it is maximal for a given domain.

4) Unavailability of information.

We clarified in the notes below the table the meaning of n/a: we did not find the information. We refer to the above comment for other modifications related to this.

5) Unexplained abbreviations.

We removed the (missing) reference to HER as we find it unnecessary for the level of details appropriate for the manuscript. In addition, we made clearer in the text that LWM refers to the cited method. We cite almost 200 papers and use the abbrevations of tens of them. While we find it unnecessary for the clarity of the text to unfold all abbrevations, it may make the text larger and more difficult to read. For this reason, we have chosen to not apply this rule regarding the names of methods. We removed the SOTA abbrevation and removed most of the abbreviations in the tables.

5) Minor comments

We fixed the minor comments. Here are some comments on your suggestions:

- Page 7 Figure 1, the trajectory does not matter as long as the agent can go to the far right part of the environment. The skill was previously learnt or given in this case, which we clarify now.
- Page 12, Table 2: we added the best score (to the best of our knowledge) and the best score using intrinsic motivation.
- Page 18 line 509: we moved the citations to Section 6.3 and added them in the table.

References:

Sekar, R., Rybkin, O., Daniilidis, K., Abbeel, P., Hafner, D., & Pathak, D. (2020, November). Planning to explore via self-supervised world models. In International Conference on Machine Learning (pp. 8583-8592). PMLR.

Mendonca, R., Rybkin, O., Daniilidis, K., Hafner, D., & Pathak, D. (2021). Discovering and achieving goals via world models. Advances in Neural Information Processing Systems, 34, 24379-24391.

Reviewer 2 Report

The paper reviews intrinsic motivation approaches from an information-theoretic perspective. The approaches are categorized into three different categories: surprise, novelty, and skill learning. In each category, the approaches are further categorized according to how they optimize the objective of their corresponding category. Overall, I find the review novel and systematically sound.   ########## Organization and Style #########

The paper is well organized. Figures and tables help the paper's comprehension.   ########## Coverage of Literature #########   1. In the skill learning category, a group of approaches [1-3] that perform skill abstraction based on a measure of competence progress in achieving sampled goals is missing.    2. In the learning progress sub-category, the focus is more on temporally local measures of learning progress. Other works [4,5] show the usefulness of spatially and temporally local learning progress using an ensemble of local forward models.   3. In the novelty maximization category, approaches that use novelty measures based on diversity [6,7] to derive intrinsic motivation are missing. Examples of such approaches are measures based on the diversity of state sequences [6] and the diversity of sampled actions to encourage actions that are less frequently sampled for learning [7]. Similarly, novelty measures that encourage behavior matching, based on the distance between demonstrated and generated trajectories in a self-learned behavior embedding space, fit into this category.   ########## Application Domains  #########   The scope of the review is well-defined and considers the use of intrinsic motivation in reinforcement learning. However, the domains of application are mostly reinforcement learning problems. I am interested to know how the reviewed intrinsic motivation paradigm can be potentially used to solve other related and interesting problems such as explainable reinforcement learning, one-shot learning, and cross-modal learning.   ############## Presentation ###############

The paper is overall well written.

Minor comments:    - Language correction needed here: "We detailed three ways to on" in Section 3. - Caption of table 2 appears above the table but it is below the table in table 3.   [1] "GRAIL: A Goal-Discovering Robotic Architecture for Intrinsically-Motivated Learning". IEEE Transactions on Cognitive and Developmental Systems8(3), pp.214-231 (2016). [2] "C-GRAIL: Autonomous reinforcement learning of multiple, context-dependent goals". IEEE Transactions on Cognitive and Developmental Systems (2022). [3] "Autonomous Reinforcement Learning of Multiple Interrelated Tasks". ICDL-EpiRob, pp. 221-227(2019). [4] "Improving robot dual-system motor learning with intrinsically motivated meta-control and latent-space experience imagination". Robotics and Autonomous Systems 133 (2020). [5] "Efficient Intrinsically Motivated Robotic Grasping with Learning-Adaptive Imagination in Latent Space". ICDL-EpiRob, pp. 240-246 (2019). [6] "Diversity-augmented intrinsic motivation for deep reinforcement learning". Neurocomputing 468 pp. 396-406 (2022). [7] "Sampling diversity driven exploration with state difference guidance". Expert Systems with Applications203, p.117418 (2022). [8] "Behavior Self-Organization Supports Task Inference for Continual Robot Learning". IROS pp. 6739-6746 (2021).

Author Response

Thank you for your suggestions. We highlighted in red our modifications to the manuscript.

1) "I am interested to know how the reviewed intrinsic motivation paradigm can be potentially used to solve other related and interesting problems such as explainable reinforcement learning, one-shot learning, and cross-modal learning."

These are interesting topics, however, we had to limit the scope of our survey and decided to focus on widely investigated issues in RL: the problem of exploration and skills learning. Thus, we leave this to future works.

2) Citations and language corrections

We added the above-mentioned citations and fixed the language issues. The notion of competence progress does not directly relate to our formalism of skill learning. But we agree it is an important intrinsic motivation in the literature and we now further discuss its relationship with time-extended prediction progress in Section 8.3.2.

Reviewer 3 Report

The authors present a sufficiently comprehensive survey of the role of intrinsic motivations (IMs) within Reinfocement Learning (RL), and in particular within Deep RL (DRL). The review is focused on the role of IMs in improving exploration, goal discovery and skill abstraction.

While the survey in itself does not add much to those already in the literature (some of which are very recent), the information-theoretic perspective presents the issues addressed from a less usual perspective that deserves to find its way into the literature. The background and definitions are detailed, as are the identification of problems and the categorisation of the articles included in the review. In some cases, the aforementioned perspective and its formalisations may confuse a reader used to think at these problems from a more RL-related perspective, but the authors were able to explain the parallels and different notations appropriately. In this perspective, I would however suggest the authors to enphasise a little bit the parallels with RL notations, especially in the section related to skill learning.

The works included in the survey constitute a proper overview of the literature tackling the issues identified by the authors. Conclusions sums up well the path traced in the article and further emphasises the open-questions of this research area.

In general, the paper is well written and I think it might give a contribution to better frame important issues within the intrinsically motivated RL research area.

As previously mentioned, I think the authors made and a proper sufficiently comprehensive selection of articles to be presented in the survey. I just want to mention some additional works that might worth a mention in relation to some of the points discussed by the authors.

With respect to goal-parametrized RL and especially in relation to autonomous intrinsically motivated learning of multiple goals, [1] provide a formalism similar to the one described by the authors in section 2.6, but it further stressed how multiple-goal learning should be described in the perspective of intrinsically motivated open-ended learning. A slightly different formalisation can also be found in [2]. On the same topic, [3] use a similar "formal approach" to learn multiple-interrelated skills.

In sec. 8.2, starting at line 761 the authors mention the fact that ther might be different critical periods when the brain is more plastic to acquire specific knowledge. On this line, and in relation to IMs, [4] underline how IMs might work not only to train skill learning, but also to signal the maturational necessity for an agent to step towards a different stage of learning.

References:

[1] Santucci, V. G., Montella, D., & Baldassarre, G. (2022). C-GRAIL: Autonomous reinforcement learning of multiple, context-dependent goals. IEEE Transactions on Cognitive and Developmental Systems.

[2] Romero, A., Baldassarre, G., Duro, R. J., & Santucci, V. G. (2022, September). Autonomous learning of multiple curricula with non-stationary interdependencies. In 2022 IEEE International Conference on Development and Learning (ICDL) (pp. 272-279). IEEE.

[3] Blaes, S., Vlastelica Pogančić, M., Zhu, J., & Martius, G. (2019). Control what you can: Intrinsically motivated task-planning agent. Advances in Neural Information Processing Systems, 32.

[4] Oudeyer, P. Y., Baranes, A., & Kaplan, F. (2013). Intrinsically motivated learning of real-world sensorimotor skills with developmental constraints. In Intrinsically motivated learning in natural and artificial systems (pp. 303-365). Springer, Berlin, Heidelberg.

Author Response

Thank you for your suggestions. We highlighted in red our modifications to the manuscript.

1) "In this perspective, I would however suggest the authors to enphasise a little bit the parallels with RL notations, especially in the section related to skill learning."

We are not sure to understand what you refer to. Please could you give precise pointers ? We tried to stick as much as possible to RL notations.

2) Related works

We clarified that "the ability of a goal-conditioned policy to generalize over different situations (irrelevant objects or light intensity)[1] depends on how g and RG are built." In this formalism, removing or keeping high-level features can be done at the level of g. We also now slightly discuss the fact that the use of expert policies may prevent generalization over the goal space.

3) Bilateral interactions between critical periods and IMs

We now mention the reference and this result where we discuss critical periods.

Round 2

Reviewer 2 Report

The comments raised were largely addressed in the revised submission. A discussion on novelty measures based on behavior-matching is still missing (ref [8] in last review). Table 1's caption should appear under the table for consistency with other tables.

Reviewer 3 Report

The authors properly improve their paper that can be now considered for publication after checking for typos.

With respect to RL notation into skill learning paragraph, I was refererring to the fact that skill learning is there described as "maximising the mutual information between the goal and the contextual states" which is totally understandable but not strictly related to RL notation (which instead is profitably introduced in other sections). Eq. 26 recall RL reward function but still leverage on information gain. Which indeed is the focus of the paper! Mine was just a suggestion to improve the parallels with a formalism (the one of RL) which in my opinion is more commonly known by people addressing similar topics.

This said, i think the paper deserves publication even in the current form so i let the authors decide if they want to follow my suggestion or not